# Mechanistic Insights into Cytokinin-Regulated Leaf Senescence in Barley: Genotype-Specific Responses in Physiology and Protein Stability

**DOI:** 10.3390/ijms26199749

**Published:** 2025-10-07

**Authors:** Ernest Skowron, Magdalena Trojak, Julia Szymkiewicz, Dominika Nawrot

**Affiliations:** Department of Environmental Biology, Jan Kochanowski University of Kielce, Uniwersytecka 7, 25-406 Kielce, Poland; magdalena.trojak@ujk.edu.pl (M.T.); julia.szymkiewicz@ujk.edu.pl (J.S.); dominika.nawrot44@gmail.com (D.N.)

**Keywords:** barley, cytokinins, benzyladenine (BA), lovastatin (LOV), leaf senescence, dark-induced senescence (DIS), stay-green phenotype, photosynthesis, reactive oxygen species (ROS), photosynthetic protein stability

## Abstract

Cytokinins (CKs) are central regulators of leaf senescence, yet their cultivar-specific functions in cereals remain insufficiently understood. Here, we examined dark-induced senescence (DIS) in three barley (*Hordeum vulgare* L.) cultivars: Carina, Lomerit, and Bursztyn, focusing on responses to exogenous benzyladenine (BA) and inhibition of endogenous CK biosynthesis via the mevalonate (MVA) pathway using lovastatin (LOV). Bursztyn, a winter cultivar, displayed a previously uncharacterized stay-green phenotype, characterized by delayed chlorophyll and protein degradation and reduced sensitivity to BA with respect to chlorophyll retention. In contrast, Carina (spring) senesced rapidly but exhibited strong responsiveness to BA. Lomerit (winter) showed an intermediate phenotype, combining moderate natural resistance to senescence with clear responsiveness to BA. CK application suppressed SAG12 cysteine protease accumulation in all cultivars, serving as a marker of senescence and N remobilization, stabilized photosystem II efficiency, preserved photosynthetic proteins, and alleviated oxidative stress without promoting excessive energy dissipation. Although BA only partially mitigated the decline in net CO_2_ assimilation, it sustained ribulose-1,5-bisphosphate regeneration, supported electron transport, and stabilized Rubisco and Rubisco activase. Moreover, LOV-based inhibition of the MVA pathway of CK biosynthesis revealed that endogenous CK contributions to senescence delay were most pronounced in Lomerit, moderate in Bursztyn, and negligible in Carina, indicating genotype-specific reliance on MVA-versus methylerythritol phosphate (MEP) pathway-derived CK pools. Collectively, these findings identify Bursztyn as a novel genetic resource for stay-green traits and demonstrate that BA delays DIS primarily by maintaining photosynthetic integrity and redox balance. The results highlight distinct regulatory networks shaping CK-mediated senescence responses in cereals, with implications for improving stress resilience and yield stability.

## 1. Introduction

Leaf senescence represents the terminal stage of plant ontogeny, preceding programmed cell death, and is regulated by a complex interplay of endogenous and environmental factors. Plant hormones play a central role in this process: ethylene (ET) and abscisic acid (ABA) act as accelerators of senescence, whereas cytokinins (CKs) are well-established inhibitors [1]. Because leaves are the primary photosynthetic organs, their senescence has a direct impact on assimilation efficiency and crop yield [2]. The process typically unfolds in three distinct phases: (i) initiation, characterized by transcriptomic reprogramming and accumulation of reactive oxygen species (ROS); (ii) degradation, involving macromolecule breakdown and nutrient remobilization; and (iii) terminal cell death [3,4].

Dark-induced senescence (DIS) constitutes a widely employed experimental model for elucidating the molecular and physiological mechanisms of leaf senescence. Darkness acts as a potent environmental cue, primarily by depleting photosynthetically derived carbohydrates, thereby accelerating senescence [5]. DIS ensures a synchronized onset of senescence across detached leaves, facilitating reproducible analyses of metabolic reprogramming. This includes early declines in chlorophyll content, photosynthetic efficiency, and protein abundance [6]. Importantly, despite its artificial induction, DIS recapitulates key features of developmental senescence, such as the activation of canonical senescence-associated genes, including *SAG12* (a cysteine protease), underscoring the conserved regulatory framework of senescence [7].

A hallmark of leaf senescence is visible yellowing [8], resulting from chlorophyll degradation and progressive dismantling of chloroplast structures [3,9]. The role of CKs in delaying this process is supported by evidence showing a decline in endogenous CK pools during senescence [10] and by transgenic approaches, such as overexpression of the *ipt* gene, which encodes isopentenyltransferase, an enzyme catalyzing a key step in CK biosynthesis, under the control of the SAG12 promoter, activated at the onset of senescence, thereby prolonging leaf greenness through a precisely autoregulated loop [11]. At the molecular level, CKs suppress chlorophyll catabolism by downregulating chlorophyll catabolic enzymes (CCEs), including NON-YELLOW COLORING 1 (*NYC1*), a chlorophyll *b* reductase that catalyzes the conversion of chlorophyll *b* (Chl *b*) into 7-hydroxymethyl chlorophyll *a* [12,13]. Inhibition of *CCE* transcription prevents Chl *b* breakdown, promoting accumulation of light-harvesting complex–chlorophyll *b* (LHC–Chl *b*) assemblies and enhancing stability of photosynthetic antenna complexes [14,15].

However, chlorophyll retention does not necessarily correspond to preserved photosynthetic efficiency, as demonstrated in so-called “stay-green” phenotypes. These include mutants or transgenic lines in which senescence is genetically altered [13]. Three major classes of stay-green mutations have been described: (i) defects in chlorophyll *b* reductase (*nyc1*, *nol*), (ii) disruption of chlorophyll catabolism via mutations in pheophorbide *a* oxygenase (*PAO*), and (iii) mutations in STAY-GREEN (SGR) proteins, which mediate disassembly of chlorophyll–protein complexes [9].

Conversely, pigment retention induced by CK application during DIS can provoke adverse effects, including photoinhibition, reduced Calvin cycle activity, diminished CO_2_ assimilation, enhanced non-photochemical quenching (NPQ), and over-reduction in PSII acceptors [16]. Furthermore, elevated ROS accumulation in CK-overproducing plants has been associated with exacerbated membrane damage [17].

The present study aims to advance the understanding of the anti-senescence effects of exogenous CKs, particularly benzyladenine (BA), in barley cultivars with contrasting senescence kinetics. Specifically, we sought to determine whether naturally occurring and BA-induced stay-green phenotypes share structural and functional features of the photosynthetic apparatus, and to what extent the rate of DIS progression is shaped by endogenous CK biosynthesis. Moreover, we assessed whether delayed senescence, either intrinsic or CK-induced, modulates oxidative stress responses. To this end, endogenous CK biosynthesis was manipulated using lovastatin (LOV), a competitive inhibitor of the mevalonate (MVA) pathway. Physiological and biochemical analyses encompassed chlorophyll *a* fluorescence (ChlF), gas exchange measurements, pigment quantification, immunodetection of key photosynthetic proteins, and assessment of oxidative stress via lipid peroxidation assays.

Barley was selected as a model species owing to its agronomic importance, well-characterized genome, and established suitability for senescence research. Among cereals, barley has become a functional analogue of *Arabidopsis thaliana* in leaf senescence studies, since Arabidopsis exhibits distinct leaf longevity and senescence programs compared with monocotyledonous crops [18,19]. Moreover, DIS in barley closely parallels natural senescence at the transcriptomic level, thus providing a robust experimental system. In cereals, senescence is intimately linked to nutrient remobilization, with leaves, particularly the flag leaf, acting as the principal source of photoassimilates and nitrogen for grain filling. Genetic regulation of this process involves transcription factors such as *Gpc-1* and *NAM-B1*, which coordinate both senescence timing and grain protein content (GPC) [20]. Elucidating the interplay between CK signaling, oxidative stress, and photosynthetic stability in stay-green phenotypes offers valuable insights into improving source–sink balance and enhancing stress resilience in cereal crops.

## 2. Results

### 2.1. Phenotypic Characterization of the Stay-Green Trait in Barley Cultivars

Based on the analysis of leaf blade yellowing following DIS treatment (Figure 1) and the evaluation of SPAD and CSI-SPAD indices, Bursztyn cultivar was selected.

SPAD index measurements revealed that cultivar Bursztyn exhibited the highest capacity to retain relative chlorophyll content under dark-induced senescence (DIS), showing only a 27% decline after 96 h (4 DDI) of treatment (Figure 2a). A delayed chlorophyll loss was also observed in cultivar Lomerit, with a 46% SPAD reduction. The calculated chlorophyll stability index (CSI-SPAD) after DIS was highest in Bursztyn (73%), followed by Lomerit (54%) and Carina (44%) (Figure 2b).

### 2.2. PSII Photochemical Efficiency Altered by Dark-Induced Senescence (DIS) and CK (BA) Treatment

Evaluation of PSII quantum efficiency revealed that under control conditions, the three barley cultivars exhibited similar maximum photochemical efficiency of PSII (Fv/Fm ≈ 0.77) (Figure 3a), but differed in effective quantum yield of PSII (ΦPSII) (Figure 3b), electron transport rate (ETR) (Figure 3d), and photochemical quenching qP (Figure 3c). Carina showed the highest ΦPSII and ETR values, indicating superior photochemical energy utilization. In contrast, Bursztyn exhibited lower values for these parameters but showed higher reduction in the primary electron acceptor Q_A_ of PSII, as indicated by 1–qP (Figure 4a). DIS induced a cultivar-dependent decline in PSII activity. Carina and Lomerit were more susceptible, showing pronounced reductions in Fv/Fm (52% and 39%, respectively), ΦPSII (62% and 44%, respectively), and ETR (62% and 53%, respectively). Bursztyn displayed greater resistance, maintaining higher qP values post-DIS, suggesting effective reactivation of electron transport following light exposure. However, all cultivars, particularly Carina, exhibited marked reductions in R_Fd_ (Figure 3e), indicating decreased photosynthetic performance (46–88%).

Treatment with 50 µM BA mitigated DIS effects across all cultivars, with the strongest protective effect observed in Bursztyn and Lomerit. BA improved ΦPSII by 14% and 5% compared to DIS alone, and enhanced qP by 6% and 23%, respectively. BA also alleviated excessive Q_A_ reduction. In the cultivar Bursztyn, BA application resulted in a 31% decrease in 1–qP compared to DIS. In Lomerit, the reduction was 16%, while in Carina, BA led to nearly a twofold decrease in 1–qP relative to the values observed under DIS incubation without BA (Figure 4a). Analysis of quantum efficiency of regulated (ΦNPQ) (Figure 4c) and non-regulated (ΦNO) (Figure 4d) energy dissipation pathways confirmed these trends. DIS treatment led to a marked decrease in ΦNPQ across all cultivars, both with and without BA. Without BA, ΦNPQ dropped by 55%, 93%, and 60% in Bursztyn, Carina, and Lomerit, respectively. BA mitigated this effect, limiting the reduction to 38%, 34%, and 31%. Similarly, BA helped maintain higher qN values after DIS (Figure 4b), especially in Lomerit (only 23% decrease), and was most effective in reducing NPQ loss in Bursztyn (41%) (Figure 4e). Regarding ΦNO, control samples showed similar values across cultivars (25–30%), but DIS alone led to sharp increases: 48%, 198%, and 145% for Bursztyn, Carina, and Lomerit, respectively. BA treatment significantly reduced these increases to 5%, 56%, and 20%, respectively.

Light response curves (0–926 µmol m^−2^ s^−1^ PPFD) further characterized cultivar-specific PSII responses (Figure 5 and Figure 6). Under control conditions, Carina (Figure 5c,d) and Lomerit (Figure 5e,f) exhibited higher ΦPSII and ETR at low to medium light intensities but were more prone to photoinhibition beyond 396 µmol m^−2^ s^−1^. Bursztyn, though less efficient at lower light levels, displayed delayed photoinhibition and higher tolerance under high light (Figure 5a,b). DIS significantly impaired light-use efficiency in Carina and Lomerit, as ETR was reduced by 89% and 81%, while Bursztyn retained partial functionality. BA treatment partially restored ΦPSII and ETR, particularly in Bursztyn and Lomerit, and shifted their curves closer to control profiles.

Finally, sustained ΦNPQ under high PPFD was critical for photoprotection during DIS. BA limited DIS-induced loss of ΦNPQ to <10% in Bursztyn (Figure 6a) and Lomerit (Figure 6e), and suppressed excess ΦNO accumulation, especially in Carina (Figure 6d), where DIS led to a 3-fold increase in ΦNO at 186 µmol m^−2^ s^−1^. Collectively, the results suggest that Bursztyn exhibits superior tolerance to DIS and benefits most from BA-induced DIS mitigation.

### 2.3. Analysis of Net Photosynthesis Rate and Its Modulation by Senescence and BA

Under control conditions, Bursztyn and Lomerit exhibited the highest net CO_2_ assimilation rate (*P*_n_) (Figure 7a) and carboxylation efficiency (*CE*) (Figure 7b), with Carina showing approximately 10–14% lower values. These differences were not attributed to stomatal conductance (*g*_s_) (Figure 7c), which remained similar among cultivars, nor to substomatal CO_2_ concentration (*C*_i_) (Figure 7d). DIS and DIS + BA treatments significantly reduced *P*_n_. Relative to the control, *P*_n_ declined by 68–78% with BA and by 75–92% after DIS alone. Bursztyn exhibited the greatest resistance to DIS and the lowest sensitivity to exogenous BA. Transpiration rate (*E*) (Figure 7e) and intrinsic water use efficiency (i*WUE*; *P*_n_/*g*_s_) (Figure 7f) also varied among cultivars. Bursztyn showed the highest *E* under control, while Bursztyn and Lomerit achieved the highest i*WUE*, indicating efficient water use for carbon gain. DIS reduced both *E* and i*WUE* across all cultivars, though BA partially mitigated this effect, especially in Lomerit, where *E* increased by ~14%. Despite substantial reductions in *g*_s_ following DIS + BA treatment, reaching up to 82% in Carina, *C*_i_ remained relatively stable, presumably due to the low light intensity during instantaneous measurements. This suggests that stomatal limitations were not the primary constraint on photosynthetic activity, thereby justifying further evaluation using light response curves to identify the true biochemical limitations of the tested cultivars during DIS. Nevertheless, *C*_i_ increased by 7% and 3% in Carina and Lomerit, respectively, following DIS.

Light-response curves (LC) show that maximum assimilation rates were observed at the plateau phase within a PPFD range of 1400–1800 µmol m^−2^ s^−1^, reaching approximately 25 µmol(CO_2_) m^−2^ s^−1^ in Bursztyn (Figure 8a) and Carina (Figure 8b), and close to 30 µmol(CO_2_) m^−2^ s^−1^ in Lomerit (Figure 8c). Mean *P*_n_ values in this range were 24.7, 24.8, and 29.9 µmol(CO_2_) m^−2^ s^−1^ for Bursztyn, Carina, and Lomerit, respectively. Following DIS incubation, mean *P*_n_ declined sharply to 7.6, 1.2, and 4.2 µmol(CO_2_) m^−2^ s^−1^. BA treatment mitigated this decline, maintaining *P*_n_ at 12.2, 5.1, and 8.2 µmol(CO_2_) m^−2^ s^−1^ in Bursztyn, Carina, and Lomerit, respectively.

Modeling of the light-response data (Table 1) revealed significant reductions in both gross (*P*_gmax_) and net photosynthesis (*P*_nmax_) under DIS, with BA alleviating these declines to some extent. For instance, *P*_nmax_ dropped by 69–94% in DIS alone, but by 50–80% in DIS + BA-treated plants. Quantum yield of CO_2_ assimilation (Φ) in the initial light range *I*_0_–*I*_comp_ (0 µmol m^−2^ s^−1^ to compensation PPFD) as well as in the next light interval *I*_comp_–*I*_200_ (compensation PPFD to 200 µmol m^−2^ s^−1^) was highest in Lomerit and showed BA-dependent retention under DIS. Still, in response to DIS, Bursztyn exhibited delayed senescence, whereas Carina showed a significantly accelerated progression. The light compensation point (*I*_comp_) at which photosynthetic CO_2_ uptake equals its respiratory release also increased significantly under DIS, especially in Carina (up to 170%), but BA reduced this increase. Dark respiration (*R*_d_), varied across cultivars and treatments. Under control, Bursztyn had the lowest *R*_d_. DIS decreased *R*_d_ in Carina and Lomerit, likely due to substrate depletion during darkness. Notably, BA preserved or even enhanced *R*_d_ after DIS, as observed in Bursztyn, where its values increased by 38% following DIS and was further elevated by BA, indicating delayed senescence and sustained respiratory activity in this cultivar.

### 2.4. Changes in Photosynthetic Protein Levels, Chlorophyll a/b Ratio, and the Senescence Marker SAG12 During Senescence and BA Treatment

The study analyzed the content of proteins involved in the regulation of CO_2_ assimilation, including the large (LSU) and small (SSU) subunits of Rubisco (Figure 9), Rubisco activase (RCA_total_) (Figure 10), as well as photosynthetic apparatus proteins associated with PSII (PsbO, PsbA, Lhcb5, Lhcb1), PSI (PsaB, Lhca4), and the senescence marker SAG12 (Figure 10). The levels of Rubisco subunits were determined by Coomassie Brilliant Blue (CBB) staining, while the remaining proteins were analyzed using Western blotting with DAB colorimetric staining. Elongation factor eEF1a was used as a loading control (Figure 10). Protein levels are presented in Table 2 as relative abundance estimated through densitometric analysis of the bands visualized on gels/membranes. For SAG12, protein accumulation was observed only in DIS-treated samples of each cultivar.

It was shown that 72 h DIS incubation affected the levels of Rubisco LSU and SSU only in a limited way. Still, the greatest decrease in LSU content was noted in the Lomerit cultivar (8% compared to control), while the highest decline in SSU was observed in Carina (11%) (Figure 9, Table 2). Application of BA limited the loss of Rubisco LSU and SSU, and in the Carina and Bursztyn cultivars even led to an increase in LSU levels above control values.

Under DIS conditions without BA, a reduction in RCA_total_ content was observed (Carina: 22%; Lomerit: 14%; Bursztyn: 26%), as well as in PSII proteins: PsbA (Carina: 24%; Bursztyn: 18%), Lhcb5 (Carina: 37%; Lomerit: 30%), Lhcb1 (Carina: 60%; Lomerit: 15%; Bursztyn: 5%) (Figure 10, Table 2). Alterations in the PsbO protein content were also observed following DIS and DIS + BA treatments, with the most pronounced effects in the Lomerit cultivar. Notably, in both Carina and Lomerit, BA treatment reduced the PsbO level compared with DIS alone, whereas in Bursztyn the opposite trend was detected. Changes in the accumulation of the PSI Lhca4 antenna during DIS displayed a restricted decomposition compared to the PSII antenna proteins Lhcb5 and Lhcb1 (Figure 10; Table 2).

Namely, for PSII antenna proteins, the strongest decline following DIS was observed in Carina and the weakest in Bursztyn. In contrast, for the PSI antenna protein both Carina and Bursztyn exhibited limited losses (9.32% and 5.32%, respectively), while a slight increase accumulation of approximately 2% was observed in Lomerit following DIS. Incubation with BA during DIS in Carina effectively mitigated the loss of RCA, PsbA, Lhcb5, and Lhcb1. In Lomerit, BA was effective in protecting RCA and Lhcb1 from decline, whereas in Bursztyn it preserved RCA, PsbA, and Lhcb1 proteins. Importantly, BA application led to an increase in the PSI antenna protein Lhca4 content (Carina: 12%; Lomerit: 15%; Bursztyn: 21%) and, to a lesser extent, the PSI core protein PsaB.

As expected, DIS treatment induced an increase in the senescence marker SAG12 across all cultivars (Figure 10, Table 2). The highest SAG12 levels were recorded in Carina, whereas the lowest were observed in Bursztyn. Interestingly, BA application attenuated SAG12 activation; thus, no signal for the SAG12 protein was detected despite DIS.

Alterations in photosystem protein content, including antenna complexes, directly influenced the accumulation of associated photosynthetic pigments. Consequently, analysis of the Chl *a*/*b* ratio showed that senescence increased the ratio by approximately 6%, 11%, and 20% in Bursztyn, Carina, and Lomerit, respectively (Figure 11). BA treatment attenuated this effect, limiting the increase to approximately 4% in Carina and approximately 6% in Lomerit. In Bursztyn, BA completely prevented the increase, maintaining the Chl *a*/*b* ratio even below the control value (13% lower).

### 2.5. Effect of the CK Biosynthetic Pathway Inhibition on Leaf Senescence

Given the significant role of CKs in inhibiting DIS-induced senescence, the differential sensitivity of barley cultivars to exogenous BA, and the observed variation in senescence onset and associated physiological and molecular responses, additional analyses were conducted to assess the contribution of endogenous CK biosynthesis via the MVA pathway, inhibited within LOV, to DIS progression. Results of chlorophyll content measurements and photographs of leaf segments from Bursztyn, Carina, and Lomerit are shown in Figure 12. In Bursztyn, Chl *a* + *b* analysis confirmed delayed senescence, reflected by reduced pigment degradation (Figure 12a,d). DIS without BA caused approximately 10% reduction in Chl *a* + *b*, while LOV further enhanced the decline to 21% relative to the control. BA addition had no protective effect on chlorophyll retention during DIS but attenuated the negative impact of LOV. In Carina, 72 h of DIS without BA reduced Chl *a* + *b* by 43% compared to the control (Figure 12b,d). Incubation with BA effectively inhibited pigment loss, limiting the decrease to less than 6%, confirming that despite its rapid senescence progression, Carina remains highly responsive to exogenous CKs. In Lomerit, DIS reduced chlorophyll content by 48% (Figure 12c,d). As in Carina, BA mitigated pigment loss, restricting the decline to about 11% under DIS. By contrast, LOV treatment caused the most severe reduction (approximately 83%), which was completely reversed by co-incubation with LOV + BA, underscoring the critical role of the cytosolic pathway of endogenous CK biosynthesis in delaying senescence in Lomerit.

Re-evaluation of chlorophyll *a* fluorescence to assess the role of MVA-derived CKs in DIS (72 h) revealed no notable differences in the effective quantum yield of PSII among cultivars under control conditions, DIS, or BA treatment (Figure 13), consistent with the previous assessment (Figure 3). By contrast, additional treatment with LOV demonstrated that LOV reduced ΦPSII by 13% in Bursztyn (Figure 13a), 38% in Carina (Figure 13b), and more than 70% in Lomerit (Figure 13c). Furthermore, co-incubation with BA (LOV + BA) substantially alleviated these effects, restricting the decline to ~4–5% in Bursztyn and Carina, and completely reversing the inhibition in Lomerit, where ΦPSII exceeded control values by nearly 11%.

### 2.6. Assessment of Oxidative Stress Induced by Senescence and BA Treatment

To evaluate the intensity of oxidative stress in leaf cells of the barley cultivars hydrogen peroxide (H_2_O_2_) content, a stable reactive oxygen species (ROS) marker was quantified spectrophotometrically following 72 h incubations in darkness with or without 50 µM BA. These values were compared to those obtained from leaf fragments incubated under light conditions (control). The analysis revealed cultivar-dependent differences in endogenous H_2_O_2_ levels under control conditions, with the lowest concentration observed in Bursztyn: 0.26 µmol H_2_O_2_ g^−1^ FW, compared to 0.35 and 0.30 µmol H_2_O_2_ g^−1^ FW in Carina and Lomerit, respectively (Figure 14a). DIS led to a marked increase in H_2_O_2_ accumulation, reaching approximately 2.5-, 3-, and 2.5-fold above control levels in Bursztyn, Carina, and Lomerit, respectively. Co-incubation with BA under DIS conditions reduced H_2_O_2_ accumulation to 1.5-, 1.8-, and 1.2-fold of the control in the same cultivars, demonstrating the effectiveness of BA in limiting ROS generation during senescence particularly in Lomerit, where BA reduced H_2_O_2_ content by half compared to the DIS condition alone.

In addition, superoxide dismutase (SOD) activity was measured in leaf fragments as an indicator of antioxidant capacity, given the distinct ROS production patterns under DIS compared to photosynthetically active leaves. The results revealed significant cultivar-specific differences. Under control conditions, Lomerit exhibited the highest SOD activity, approximately 1.3-fold and 1.7-fold higher than that of Bursztyn and Carina, respectively (Figure 14b). DIS resulted in reduced SOD activity by 22%, 37%, and 46% in Bursztyn, Carina, and Lomerit, respectively. The effect of BA on SOD activity varied among cultivars. In Bursztyn, BA had no protective effect; SOD activity decreased similarly to the DIS-only condition. In Carina, BA treatment stimulated SOD activity, which was 22% higher than the control following incubation. In Lomerit, BA effectively mitigated the decline in SOD activity, maintaining it at levels comparable to the control. These findings confirm the differential sensitivity of the tested barley cultivars to exogenous CKs, with Bursztyn showing low responsiveness in the case of SOD.

To assess membrane damage during DIS-induced senescence, the content of TBARS (thiobarbituric acid-reactive substances), including malondialdehyde (MDA), a byproduct of lipid peroxidation was analyzed. Under control conditions, Lomerit exhibited the lowest level of lipid peroxidation, approximately 2.2-fold and 2.8-fold lower than in Bursztyn and Carina, respectively (Figure 14c), correlating with higher SOD activity. Incubation in darkness resulted in a substantial increase in membrane damage, reaching 58%, 65%, and 233% above control levels in Bursztyn, Carina, and Lomerit, respectively.

## 3. Discussion

Previous studies have shown that CKs, such as BA, induce swollen chloroplasts enriched in plastoglobules (PGs), peripheral grana displacement, and the accumulation of large starch grains, reflecting sustained sink-like activity of the leaf and elevated Rubisco content, partly mediated by cell wall invertase (CWINV) activity [21]. CKs also promote thylakoid hyperstacking [22] and, in senescing leaves, restore photosynthetically competent plastids by facilitating the re-differentiation of degraded grana and stroma [23].

At the molecular level, CKs repress chlorophyll catabolic enzyme (*CCE*) genes, blocking Chl *b* turnover and stabilizing LHC–Chl *b* complexes. This repression extends to key CCEs (*NYC1*, *NOL*, *PPH*, *PAO*, *RCCR*), which normally cooperate in the SGR–CCE–LHCII complex to dismantle pigment–protein assemblies during senescence [12,24]. By interfering with this pathway, CKs maintain granal compaction, delay pigment loss, and preserve photosynthetic capacity [14]. Together, these processes highlight the central role of CKs as antagonists of senescence and provide a mechanistic basis for the stay-green phenotype (Figure 15).

### 3.1. Monitoring Barley Cultivars for Stay-Green Phenotype and Sensitivity to CK-Mediated Delay of Leaf Senescence

Comparative analysis of barley cultivars identified the winter cultivar Bursztyn as exhibiting delayed leaf senescence during DIS (Figure 1). Leaf senescence progression, based on SPAD value changes and modeled using the Gompertz curve, can be divided into two main phases [25,26]. The initial phase involves a gradual loss of chlorophyll of no more than 10% relative to the original level and is considered the onset of senescence [26]. This is followed by a biphasic second stage, during which chlorophyll content first declines from 90% to approximately 70%, and then decreases rapidly to around 30%. The interval between 70% and 30% represents the most accelerated stage of leaf senescence [25]. Consequently, after 96 h (4 DDI) of DIS incubation, Bursztyn, and to a lesser extent Lomerit, exhibited reduced chlorophyll degradation, amounting to 73% and 54%, respectively, whereas Carina showed a more moderate decline, retaining approximately 30% of the initial chlorophyll content (Figure 1 and Figure 2). Notably, BA application during 72 h of DIS incubation did not significantly affect chlorophyll retention in the Bursztyn cultivar, suggesting both stay-green phenotype traits and low CK sensitivity (Figure 12a). In contrast, following DIS incubation, leaf senescence in Carina and Lomerit entered the second phase of rapid chlorophyll degradation (Figure 2a,b and Figure 12b,c), and application of 50 μM BA slowed down this process, preventing the transition into the final stage of chlorophyll loss. Although 50 μM BA was less effective in delaying senescence in Lomerit, this cultivar exhibited a naturally slower senescence rate compared to Carina, as demonstrated after 96 h of DIS without BA treatment (Figure 2b). The results are promising, since unlike Carina and Lomerit, which are well characterized with respect to senescence-related traits [27,28], Bursztyn has not previously been investigated in the context of senescence or stay-green characteristics, and the mechanisms underlying its delayed DIS-induced senescence remain largely unknown.

However, a previous analysis compared the pattern of leaf senescence in the stay-green mutant (*N22-H-dgl162*) of the Nagina22 (N22) rice cultivar with that of BA-treated N22 plants [13]. The authors reported that the expression of chlorophyllide *a* oxygenase (*CAO*) increased within 24 h of BA treatment under DIS and remained elevated up to 72 h. In addition, chlorophyll *b* reductase (*NOL*, NYC1-like) and hydroxymethyl chlorophyll *a* reductase (*HCAR*) were strongly induced after 72 h under BA treatment during DIS, whereas their expression levels were reduced in untreated samples. CAO catalyzes the oxidation of Chl *a* to Chl *b*, while NOL and HCAR catalyze the reverse reaction through the shared intermediate 7-hydroxymethyl chlorophyll *a* (HmChl *a*). Collectively, CK-mediated inhibition of Chl *b* degradation and the partial redirection of the Chl *a* pool toward HmChl *a* contribute to the stabilization of light-harvesting complexes (LHCs) and delay the onset of senescence [13]. Thus, the mode of action of both exogenous CK application and stay-green phenotypes appears to be closely associated with chlorophyll retention. Importantly, future analyses of the barley cultivar Bursztyn should also address changes in the expression of chlorophyll metabolism-related genes as well as the dynamics of HmChl *a* accumulation, especially since total Chl *a* + *b* content was maintained at comparable levels in BA-treated and untreated samples under DIS (Figure 12).

### 3.2. Chlorophyll a Fluorescence Analysis

ChlF analysis confirmed that DIS led to reduced PSII activity, both maximal and effective, across all cultivars. The decline was most pronounced in Carina, as indicated by Fv/Fm, ΦPSII and ETRII measurements, consistent with earlier findings showing accelerated PSII photochemical decline during senescence in this cultivar [29]. In contrast, Lomerit maintained more stable PSII function, in line with previous reports linking synchronized PSI and PSII deactivation with delayed photosynthetic decline [27]. Although Bursztyn displayed lower baseline ΦPSII under control conditions (Figure 3b), it showed minimal reductions in PSII efficiency and electron transport rates during DIS. Moreover, the rapid light response curve (RLC) indicates that only Bursztyn retained near-control photochemical activity under DIS, supporting its stay-green phenotype (Figure 5a,b). However, a notable reduction in R_Fd_ suggests that despite preserved light-phase activity, CO_2_ assimilation efficiency was impaired (Figure 3e). BA treatment exerted a protective effect on PSII across all cultivars. In Bursztyn and Lomerit, BA maintained effective PSII activity above control levels, while Carina responded less favorably. R_Fd_ values were also better preserved by BA, particularly in Bursztyn and Lomerit.

Analysis of non-photochemical quenching parameters (Figure 4) revealed PSII inhibition in Carina post-DIS, as indicated by elevated 1–qP values (Figure 4a). In Bursztyn, 1–qP remained largely unaffected, while in Lomerit it increased moderately, yet with maintained regulated energy dissipation, which was nearly absent in Carina (Figure 4b). BA reduced Q_A_ reduction levels below or to control levels in all cultivars, aligning with previous findings on CK-mediated Q_A_ oxidation in darkness [30]. DIS increased passive energy dissipation (ΦNO), especially in Carina, whereas Bursztyn showed the smallest increase (Figure 4d). qN analysis reflected trends similar to NPQ but offered greater sensitivity, detecting initial stages of non-photochemical quenching before full thermal dissipation activation [31]. DIS limited thermal quenching more than membrane energization, with Carina showing near-complete inhibition of both qN and NPQ (Figure 4e).

Under higher PPFD, Carina exhibited delayed NPQ induction (Figure 6c), suggesting involvement of slower-acting components beyond qE [32,33]. Comparison of energy distribution under BA–DIS conditions showed that in Bursztyn and Lomerit, BA preserved photochemical efficiency by reducing ΦNPQ without strongly increasing ΦNO (Figure 6). Though less effective in Carina, BA still enhanced regulated dissipation and reduced passive energy loss, ultimately improving PSII photochemical performance. At higher PPFD, BA promoted NPQ induction to control levels in Bursztyn and Lomerit, and to levels exceeding DIS in Carina.

### 3.3. Leaf Gas Exchange Under Dark-Induced Senescence and BA Treatment

Consistent with a previous study [30], dark-induced senescence markedly reduced CO_2_ assimilation and *g*_s_ across all cultivars, without substantially affecting *C*_i_ (Figure 7). BA application only partially mitigated CO_2_ assimilation decline, notably in Carina and Lomerit. At chamber-specific low light (PPFD 130 μmol m^−2^ s^−1^), *P*_n_, *CE*, and *g*_s_ did not differ substantially among cultivars. At the same time, however, results revealed that photosynthetic limitation in response to DIS was governed primarily by carboxylation rate rather than CO_2_ diffusion, as the *C*_i_ remained high despite reduced *g*_s_. Consequently, DIS induced a significant *CE* decrease, especially in Carina, while BA application slightly mitigated the *CE* reduction. In contrast, BA failed to prevent DIS-induced stomatal closure, aligning with the limited CK effect previously reported [34]. The possible explanation for reduced *g*_s_ during DIS is related to the depletion of chloroplastic starch reserves in guard and mesophyll cells in response to prolonged light deprivation. Consequently, the light-driven osmotic opening of stomata in response to light is impaired [35].

Furthermore, to identify other limiting factors during DIS and to evaluate the effects of CK on CO_2_ assimilation, photosynthetic performance was further assessed under saturating PPFD within the light response curve. LC analyses showed that under control conditions, *P*_nmax_ indicated no enhancement of CO_2_ fixation in Bursztyn compared with Carina, whereas Lomerit exhibited superior photosynthetic performance [27]. At the same time, DIS incubation reduced *P*_nmax_ by approximately 3-fold in Bursztyn (twofold with BA), 7-fold in Lomerit (3.5-fold with BA), and 17.5-fold in Carina (5-fold with BA). Actual CO_2_ fixation was inhibited more strongly than suggested by chlorophyll fluorescence parameters, such as the fluorescence decrease ratio R_Fd_, although Bursztyn retained relatively high Calvin cycle activity after DIS.

Previous studies have documented that CKs modulated dark respiration (*R*_d_), and delayed senescence by repressing polysaccharide catabolism and enhancing mitochondrial coupling, conserving respiratory substrates during prolonged light deprivation [34,36]. In such a scenario, the mode of action of CKs is associated with reduced respiration rate, postponing the onset of the senescence process due to the exhaustion of starch. In our study, while BA did not fully sustain *R*_d_ at control levels, the respiration rate after 96 h of DIS remained significantly higher relative to CK-free treatments. Without BA, the *R*_d_ declined in Carina and Lomerit during DIS, reflecting substrate depletion. In contrast, Bursztyn, characterized by an inherently lower baseline of *R*_d_, maintained higher post-DIS respiration, even without BA application, presumably because of a higher CK intrinsic level. These findings accord with previous evidence that CKs stabilize ATP levels during prolonged darkness by suppressing respiration [37] and delaying the peak of respiration-associated chlorophyll loss [36]. Starch reserves, critical for sustaining metabolism during darkness [38], when preserved, prevent premature carbohydrate starvation and autophagy [39]. The observed *R*_d_ patterns suggest that endogenous CK levels and starch turnover dynamics jointly underpin cultivar-specific resilience to DIS. Whether similar mechanisms modulate developmental senescence remains unresolved; while soluble sugar accumulation can accelerate senescence [40,41], its causal role is debated [7]. Notably, starch-derived sugars promote Rubisco-containing body (RCB) formation during DIS, paralleling early events of physiological senescence [42].

### 3.4. Stability of Rubisco, Rubisco Activase, and Photosynthetic Proteins During Senescence and in Response to BA Treatment

Incubation with BA limited the degradation of both Rubisco subunits, most prominently in Bursztyn. DIS reduced the efficiency of Rubisco activation by RCA by ~25% in Bursztyn, ~20% in Carina, and ~15% in Lomerit relative to controls. Pre-treatment with BA protected RCA levels in all cultivars and fully preserved them in Bursztyn. Both *α*- and *β*-RCA isoforms characteristic of barley were detected, with *β* prevailing under all treatments. Previous studies indicate distinct roles of these isoforms, with *α* linked to stress acclimation and *β* to Rubisco regulation under stable conditions [43]. Our findings, consistent with earlier reports [13], suggest that protein loss during DIS is driven mainly by enhanced degradation rather than reduced synthesis, with RCA degraded earlier than Rubisco. Thus, the protective effect of BA likely reflects the inhibition of the protein degradation.

Furthermore, previous studies have shown that leaf senescence is accompanied by sequential degradation of photosynthetic proteins. In barley, D1 protein (PsbA) is lost first, followed by cyt*f* and RbcL. Subsequently, mRNA levels of *psbA* and *petC* decline, and proteins such as cyt*b*_559_ and RbcS degrade, with LHCII being among the last to be affected [29]. A detailed analysis in *Arabidopsis thaliana* revealed a similar pattern: early degradation of D1, Lhcb3, Lhca1, Lhca3, cyt*b*_6_*f*, and RbcL, while proteins such as Lhcb1, Lhcb2, PsaA/B, Lhca2, and ATP synthase remained more stable throughout the senescence process [44].

Western blots revealed that, in the cultivar Carina, DIS led to an early and pronounced degradation of PSII antenna proteins, particularly those of the LHCII complex and Lhcb5 (CP26), followed by degradation of the PSII core protein PsbA. In PSI, a strong reduction in core protein PsaB and a milder degradation of antenna protein Lhca4 were observed. The order of protein degradation in Carina under DIS was as follows: Lhcb1 > Lhcb5 > PsaB > PsbA > Lhca4. BA incubation mainly protected PSI antennae in Carina, with higher protein levels than in the light-incubated control. For PSII, BA provided stronger protection to the core proteins. In the Lomerit cultivar, DIS also triggered degradation of PSII antenna proteins, especially Lhcb5, with minimal impact on core PSII proteins and LHCI. Overall, the sequence of protein degradation proceeded as follows: Lhcb5 > PsaB > Lhcb1 > Lhca4 > PsbA. BA treatment protected mainly Lhcb1 and Lhca4. For the delayed-senescence cultivar Bursztyn, DIS mainly affected core proteins PsbA more severely than PsaB while antenna proteins remained relatively stable. BA significantly enhanced the accumulation of Lhcb1 (~25%), Lhca4 (~21%), and PsaB (~8%) compared to the control. BA also slightly preserved PsbA levels, but not Lhcb5. The degradation order during DIS in Bursztyn was: PsbA > PsaB > Lhcb1 > Lhca4 > Lhcb5.

Previous studies in barley (Golden Promise) showed early degradation of Lhcb1, PsbA, and Lhcb4 during senescence, while PSI proteins like PsaA and Lhca1 remained stable [45]. In another study, PSI center was more sensitive to senescence than PSII [46]. Still, however, both studies confirm high stability of LHCI proteins during leaf senescence. The role of CKs in protecting PSII and PSI structures has been demonstrated by another study [47], showing that BA protects LHCII and PSII cores during DIS, possibly by stabilizing their structure and slowing Chl *b* loss. Our study supports this, showing preferential BA-mediated protection of antennae in Bursztyn and Lomerit, and PSI antennae in Carina.

Protein degradation is associated with pigment loss. An increased Chl *a*/*b* ratio under DIS indicated preferential degradation of Chl *b*-containing structures like LHCII, suggesting grana disassembly [27]. This is cultivar-dependent: in Carina, DIS caused an increase in Chl *a*/*b* and significant Lhcb1 loss, indicating early LHCII degradation. In contrast, Bursztyn showed only a minor reduction in Lhcb1, suggesting either degradation of other antenna proteins or impaired chlorophyll *a* catabolism, for example, due to reduced activity of enzymes such as PAO. This may explain the increased Chl *a*/*b* ratio and, presumably, the stay-green phenotype [13,48]. BA incubation reversed the DIS-induced Chl *a*/*b* increase in Bursztyn, likely by stabilizing Lhcb1, preserving Chl *b* and/or PAO activity. Photosynthetic performance analyses confirmed that PSI efficiency declined under DIS in both Carina and Lomerit, mainly due to PsaB loss. PSII activity decreased more severely in Carina, as blots showed major losses in Lhcb1, Lhcb5, and PsbA. In Lomerit, PSII decline was due to Lhcb5 degradation.

Furthermore, NPQ was significantly suppressed in Carina under DIS, likely due to LHCII loss [49]. Lomerit maintained higher Lhcb1 levels, suggesting antenna dissociation from PSII rather than degradation. This was supported by increased NPQ, consistent with LHCII detachment and grana loosening [50]. Loss of Lhcb5, a linker protein, may explain the dissociation [51]. In Carina, BA treatment preserved PSI proteins, prevented PsbA degradation, but did not stop LHCII loss. This might reduce energy transfer to PSII, enabling PSI protection and D1 repair via cyclic electron flow (CEF) and ATP production. Grana disassembly is essential for D1 repair [50]. CK signaling mutants show impaired D1 recovery under light stress [52].

DIS did not significantly affect PsbO levels. Where reductions occurred (e.g., Lomerit), BA had little protective effect, suggesting CKs do not directly regulate OEC stability. PsbO remained stable in Carina and Bursztyn, indicating delayed degradation despite senescence [53]. Similar PsbO stability under drought was reported previously [54]. In summary, BA protected photosystems by preserving antenna structures, minimizing unnecessary protein degradation, and enabling more effective photoprotective responses upon light re-exposure.

### 3.5. Effect of BA on the Accumulation of the Senescence Marker SAG12 During DIS

SAG12 belongs to the cathepsin L-like family, subgroup A [55], and is a senescence-associated protease and a specific molecular marker of leaf senescence [56]. It is typically activated during the late stages of this process [57], and its accumulation is closely correlated with CK levels [58]. SAG12 levels have previously been found to increase in barley leaves during senescence, as this protease is involved in Rubisco degradation. However, it should be emphasized that among the barley papain-like cysteine proteases (PLCPs), HvPAP-17 is an ortholog most closely related to the Arabidopsis SAG12 gene, sharing approximately 53% sequence identity, which explains its cross-reactivity with the anti-SAG12 antibody [55]. Although some authors have referred to *HvPAP-17* as *HvSAG12* [59], both genes have been shown to be induced in response to 3 DDI [60].

Our results confirmed that mechanical detachment of leaves followed by incubation under light with a maintained photoperiod regime did not induce senescence to an extent sufficient to activate the SAG12 protease gene and protein accumulation. However, it should be noted that leaf detachment serves as a factor accelerating SAG12 activation when senescence is subsequently induced by DIS. This has been demonstrated in previous study [53] that reported minimal SAG12 expression in intact leaves after three days of DIS, with a significant increase only after ten days of treatment. Thus, leaves detachment clearly accelerates leaf senescence in response to DIS. Interestingly, in the present study, SAG12 accumulation was not detected in either control groups or in BA-treated samples subjected to DIS, across all cultivars, despite detachment. In contrast, SAG12 was detected after DIS incubation without BA, with the highest accumulation observed in the Carina cultivar and the lowest in Bursztyn. This finding indicates that, despite the pronounced delay in visible senescence in Bursztyn, senescence-associated proteins are still activated at the molecular level. Similar results were obtained by other study [47], which demonstrated that SAG12 expression is induced both in wild-type Golden Promise barley and in the delayed-senescence WHIRLY1 RNAi knockdown line W1-7, albeit at lower levels than in untreated controls, consistent with the reduced SAG12 accumulation observed in Bursztyn. Other authors [61] also reported comparable outcomes, showing that four days of dark incubation of detached Arabidopsis leaves led to substantial SAG12 transcript accumulation, whereas no such induction was detected in light-incubated controls. The induction of SAG12 gene expression under DIS [62], showed the first peak of SAG12 accumulation on the third day of DIS.

### 3.6. The Role of Inhibited Endogenous CK Synthesis in Cultivars with Differing Senescence Dynamics

In this study, the biosynthesis of endogenous CKs was modulated through inhibition of the MVA pathway using LOV, which blocks the synthesis of the CK precursor dimethylallyl pyrophosphate (DMAPP) [63,64,65]. LOV is a competitive inhibitor of 3-hydroxy-3-methylglutaryl-coenzyme A reductase (HMGR, EC 1.1.1.34) [66], an enzyme involved in the synthesis of endogenous CKs via the MVA pathway, thereby suppressing or limiting the formation of mevalonic acid, which is converted to DMAPP. DMAPP is then attached to ATP by the enzyme isopentenyltransferase (IPT), resulting in the formation of *N*^6^-isopentenyladenine ribotides. Subsequently, hydroxylation of the isoprenoid side chain by cytochrome P450, followed by the conversion of ribonucleosides to free bases catalyzed by LONELY GUY (LOG), produces active CKs [67]. Moreover, in the case of the plant enzyme IPT, DMAPP is almost exclusively utilized as the prenyl donor, highlighting the crucial role of the MVA pathway in the biosynthesis of endogenous CKs [68]. This approach avoids the non-specific effects associated with inhibiting the alternative, plastidial methylerythritol phosphate (MEP) pathway, which is involved in the synthesis of CKs but also carotenoids, chlorophyll side chains, plastoquinone, phylloquinone, gibberellins, and ABA [69]. While MVA inhibition also affects sesquiterpenes, triterpenes, dolichols, and brassinosteroids [70], it specifically reduces CK levels shortly after application (10–30 min), resulting in suppressed cell division and accelerated leaf senescence [71,72,73,74].

Experimental application of low concentrations of LOV (40 μM) significantly enhanced chlorophyll loss in the cultivar Lomerit, had a moderate effect in Bursztyn, and had no significant impact in Carina. Co-application with exogenous BA partially or fully mitigated these effects [73,75]. The differential chlorophyll retention LOV + BA and BA treatments suggest genotype-specific contributions of MVA-controlled endogenous CK biosynthesis to anti-senescence responses, as well as to the pattern of leaf senescence progression under DIS.

Photosynthetic analyses mirrored chlorophyll data in Lomerit and Bursztyn, while Carina showed a distinct pattern. In Bursztyn, LOV reduced PSII with incomplete reversal by BA, whereas in Lomerit, LOV clearly accelerated senescence and reduced photosynthetic activity, fully rescued by LOV + BA. Conversely, in Carina, LOV improved ΦPSII compared to DIS, suggesting delayed senescence, presumably due to compensation via the MEP pathway. This supports the hypothesis that MEP dominates CK synthesis in Carina. Inhibiting MVA may upregulate MEP activity and/or HMGR expression, bypassing the MVA block [69,72,76]. Additionally, MEP-derived isopentenyl diphosphate (IPP) and DMAPP may supplement cytosolic CK biosynthesis [63], mitigating LOV effects. Thus, Carina’s rapid senescence upon DIS induction may stem from chloroplast-localized CK synthesis being disrupted during plastid degradation, while Lomerit maintains higher CK production via MVA, delaying senescence despite chloroplast loss under DIS.

Paradoxically, MVA inhibition in Carina may enhance CK synthesis via MEP, exceeding basal levels and slowing senescence. This may also explain Carina’s strong sensitivity to exogenous CKs, which boost endogenous CK pools preferentially in chloroplasts, further suppressing senescence. Bursztyn’s moderate response to LOV and lack of accelerated senescence despite MVA inhibition suggest its stay-green phenotype is not primarily MVA-dependent, though MEP-derived CKs may still play a compensatory role. Previous analyses of tobacco cultivars differing in their leaf senescence patterns [77] demonstrated that cv. Xanthi, which exhibits a delayed onset of chlorophyll degradation under DIS, was mostly unresponsive to LOV. In contrast, in the rapidly senescing cv. Monte Calme Yellow, LOV accelerated its progression, whereas BA effectively inhibited leaf yellowing. Interestingly, the cited authors also evaluated clomazone, an inhibitor of the alternative, MEP pathway of CK biosynthesis, but it proved less effective than LOV in promoting yellowing.

### 3.7. Oxidative Stress Intensity Following DIS and BA Treatment

H_2_O_2_ is produced during electron transport, particularly under stress conditions impairing photosystems, and during dark-induced senescence, where mitochondria and peroxisomes become the main ROS sources [28,78]. H_2_O_2_ levels are closely linked to SOD activity, which catalyzes the dismutation of superoxide radicals into H_2_O_2_, subsequently scavenged by downstream antioxidant enzymes [79,80]. In this study, a decrease in SOD activity was observed across all cultivars under DIS conditions, accompanied by increased H_2_O_2_ accumulation. This pattern aligns with previous findings indicating that prolonged oxidative stress and H_2_O_2_ accumulation during senescence may suppress SOD gene expression, leading to enhanced lipid peroxidation [28,81], suggesting that the elevated H_2_O_2_ levels were not due to SOD activity itself. Moreover, lipid degradation and ROS generation have been identified as notable aspects of DIS, as evidenced by increased lipid peroxidation in cell membranes and associated alterations in membrane permeability. Furthermore, the accumulation of membrane lipid peroxidation by-products preceded the decline in chlorophyll and protein abundance [82].

BA reduced H_2_O_2_ accumulation in all cultivars, likely by restricting the rapid increase in mitochondrial respiration under prolonged darkness, a key source of ROS during DIS [36]. This reduction paralleled lower TBARS levels, a marker of lipid peroxidation [83]. Under control conditions, the cultivars displayed clear differences in SOD activity and lipid peroxidation rate, with Lomerit showing the highest SOD activity and lowest TBARS content, while Carina exhibited the opposite pattern. This reflects their distinct senescence rates induced by leaf detachment, and possibly different levels of endogenous CK, presumably higher in the slower-senescing Lomerit. Bursztyn, characterized by delayed senescence, displayed intermediate SOD and TBARS levels, suggesting a moderately efficient antioxidant defense compared to Lomerit. The differential response to BA treatment further supports the cultivar-specific sensitivity to exogenous CKs. In Bursztyn, BA had negligible effect on maintaining SOD activity or limiting TBARS accumulation, which may be due to already low oxidative stress levels and slower senescence progression. In contrast, in Lomerit, BA effectively preserved SOD activity and reduced TBARS levels relative to DIS alone. Notably, in Carina despite its low baseline SOD activity, BA enhanced antioxidant capacity above control levels and restored TBARS content to near-control values, highlighting its high responsiveness to exogenous CKs.

## 4. Materials and Methods

### 4.1. Plant Material and Growth Conditions

Barley cultivars Carina and Lomerit were selected for analysis based on previous research data [27,28], while Bursztyn was selected within the DIS screening test based on the rate of their senescence progression within SPAD analyzer (Figure 1 and Figure 2). Then, selected barley seeds of Bursztyn (winter cv.), Carina (spring cv.) and Lomerit (winter cv.) were germinated on moist paper for 2 days at 4 °C in darkness, followed by 2 days at 21 °C. Uniform seedlings were planted in seedling trays filled with organic substrate (pH 5.5–6.0) composed of deacidified peat, perlite, silica, and slow-release fertilizer (Scotts Poland, Warsaw, Poland). Growth conditions: day/night temperature 23 ± 1 °C/20 ± 1 °C; 16/8 h photoperiod; PPFD 130 μmol m^−2^ s^−1^ provided by PX256 PxCrop LED lamps (PXM, Podleze, Poland). LED spectrum: red (671 nm, FWHM 25 nm), green (524 nm, FWHM 40 nm), blue (438 nm, FWHM 20 nm) in a 9:9:8 ratio, measured with GL SPECTIS 5.0 Touch (GL Optic, Weilheim/Teck, Germany). Atmospheric CO_2_ 420 ± 20 µmol mol^−1^ and relative humidity 50–60%. Plants were watered as needed with tap water. Analyses were performed on the first true leaf at 14 DAS (days after sowing, Zadoks stage 13) [84], with two fully developed and one emerging leaf. Available data indicate that Bursztyn is medium-to-low yield performance cultivar, Lomerit is a high-yielding winter, whereas Carina rather medium-yield spring cultivar [27,85,86].

### 4.2. Leaf Senescence Induction and Chemical Treatment

Leaf samples were collected between 08:00 and 10:00 AM to minimize diurnal variation in gene and protein expression [87]. From the first fully expanded leaf, 50 mm segments were excised after removing 15–20 mm from both ends. Samples were placed adaxial side up in 85 mm Petri dishes containing 15 mL of distilled water (H_2_O_d_) with 0.2% DMSO (Honeywell International Inc., Charlotte, NC, USA) [88], supplemented with either 50 µM benzyladenine (BA) (Sigma-Aldrich, St. Louis, MI, USA) or 40 µM LOV (Sigma-Aldrich), based on previous protocols [71,73,89]. Six leaf segments were placed per dish to allow free floating. Dishes were sealed with 90 mm lids and grouped by treatment. Control samples (C) were incubated in H_2_O_d_ under growth light conditions for 72 h [59,87], while BA- or LOV-treated samples were kept in darkness. Senescence controls (DIS) were incubated in water in complete darkness. Samples were gently mixed daily without exposing them to light. After incubation, leaves designated for biochemical assays were blotted dry, flash-frozen in liquid nitrogen for ≥30 min, and stored at −80 °C [25]. For gas exchange measurements, leaves remained attached to the plant. Senescence was induced by covering selected leaves with a light-impermeable, breathable fabric for 96 h. Prior to this, leaves were sprayed twice (3 h apart, between 07:00 and 10:00 AM) with either distilled water (C, DIS) or 50 µM BA (BA treatment), at 1 mL per leaf per application, following a previous protocol [29]. All solutions were freshly prepared and contained 0.01% Tween 20 (Bio-Rad, Hercules, CA, USA) to enhance foliar uptake [73,90]. Both experimental approaches, detached leaves kept in darkness and individually darkened attached leaves under normal photoperiodic conditions, are widely accepted models for studying dark-induced senescence [18,91].

### 4.3. Determination of Photosynthetic Pigment Content and Chlorophyll Stability Index

Leaf samples were ground to a fine powder in liquid nitrogen, and chlorophyll *a*, chlorophyll *b* (Chl *a*, *b*), and carotenoids were extracted using dimethyl sulfoxide (DMSO) at a ratio of 1.5 mL DMSO per 10 mg of tissue. The extracts were incubated at 65 °C for 3 h, and absorbance was measured at 480, 649, and 665 nm using a microplate spectrophotometer (Mobi, MicroDigital Co., Ltd., Seongnam, Republic of Korea) with six replicates [92,93].

Relative chlorophyll content in 50 mm barley leaf segments was estimated non-invasively using a SPAD-502 chlorophyll meter (Minolta Camera Co. Ltd., Osaka, Japan). Measurements were taken at three positions along each segment, averaged, and repeated after incubation. SPAD values were recorded between 08:00 and 10:00 before treatment and after 96 h incubation under control (light) or DIS conditions.

### 4.4. Chlorophyll Fluorescence Kinetics and Light Response Curve

Chlorophyll fluorescence (ChF) induction in detached barley leaves was measured using a PAM fluorometer (Maxi IMAGING-PAM M-Series, Walz, Germany). Minimal fluorescence (Fo) was recorded under low-intensity modulated blue light (λ = 450 nm, 0.01 µmol m^−2^ s^−1^), while maximal fluorescence in the dark-adapted state (Fm) was induced by a 0.8 s saturating pulse (2700 µmol m^−2^ s^−1^) following 30 min of dark adaptation. Leaves were then illuminated with blue actinic light (186 µmol m^−2^ s^−1^), and after 4 min, steady-state fluorescence (Fs) and maximal fluorescence in the light-adapted state (Fm′) were recorded. Based on these values, Fv/Fm, ΦPSII, ETR, R_Fd_, qP, ΦNPQ, ΦNO, NPQ, and qN were calculated [94]. Excitation pressure on PSII (1–qP) was estimated as described earlier [95] (Table 3).

A 400 s rapid light response curve (RLC) was conducted on pre-illuminated leaves using six 20 s steps of increasing light intensity (0, 56, 186, 396, 611, 926 µmol m^−2^ s^−1^). The initial point (0 μmol m^−2^ s^−1^) reflected light-adapted fluorescence and differed from dark-adapted measurements. The curve allowed estimation of maximum PSII activity, saturation irradiance, and energy partitioning between photochemical and non-photochemical quenching pathways. Changes in ΦPSII, ETR, ΦNPQ and ΦNO were analyzed as a function of light intensity. Measurements were performed on the detached fragments of first fully expanded leaves, between 08:00 and 11:00, with six replicates per treatment.

### 4.5. Leaf Gas Exchange Measurements

Photosynthetic parameters were measured using a LI-6400XT Portable Photosynthesis System (LI-COR Inc., Lincoln, NE, USA) equipped with a 2 × 3 cm transparent chamber (6400-08) for instantaneous gas exchange or a red-blue light module (6400-02B; 665 ± 10 nm and 470 ± 10 nm) for light response curve (LC) assessment. Each treatment included twelve replicates for instantaneous measurements and six for LC analysis. Chamber conditions were standardized at 60% relative humidity, 23 °C, external CO_2_ concentration of 400 µmol mol^−1^, and a gas flow rate of 500 ± 2 µmol s^−^^1^. Actual leaf assimilation area was determined post-measurement via high-resolution leaf scans analyzed in AxioVision 4.8 (Carl Zeiss Inc., Oberkochen, Germany). Instantaneous measurements included net photosynthetic rate (*P*_n_), carboxylation efficiency (*CE*), the intercellular CO_2_ concentration (Ci), stomatal conductance (*g*_s_), and transpiration rate (*E*), stabilized within 300 s per replicate. Water-use efficiency was assessed as intrinsic WUE (i*WUE* = *P*_n_/*g*_s_) following a previous protocol, as it closely correlated with whole-plant *WUE* [102].

LCs were generated across a PPFD range of 0–2000 μmol m^−2^ s^−1^ after 20 min pre-illumination at 1500 μmol m^−2^ s^−1^ to ensure acclimation. Following stabilization of CO_2_ (20–30 s), a rapid light curve protocol was initiated [103,104] with descending PPFD steps: 2000 to 0 μmol m^−2^ s^−1^. Each step lasted 120–200 s. Then, photosynthetic parameters were estimated by fitting measured data to a nonlinear regression model [105] using Excel’s Solver function. The fitted model, based on minimizing the sum of squared errors (SSE), provided values for gross (*P*_gmax_) and net (*P*_nmax_) maximum photosynthesis, dark respiration rate (*R*_d_), light compensation point (*I*_comp_), and quantum efficiencies of CO_2_ fixation within two light ranges: Φ*I*_0_–*I*_comp_ and Φ *I*_comp_–I_200_, reflecting initial light-use efficiency [104].

### 4.6. Leaf Protein Extraction and Densitometric Quantification

Soluble leaf proteins (SLPs) were extracted using Plant Total Protein Extraction Kit (Sigma-Aldrich) according to the manufacturer’s instructions. In brief, 200 mg of the leaf powder was washed with methanol working solution and acetone, protected from proteolysis with protease inhibitor cocktail. Purified tissue pellet has been used for total protein extraction with chaotropic protein reagent. Protein content was estimated via absorbance at 280 nm using a NanoDrop 2000 spectrophotometer (Thermo Fisher Scientific, Waltham, MA, USA). Then, each extract, mixed with Laemmli Sample Buffer (Bio-Rad, Hercules, CA, USA), was loaded onto 4–20% gradient TGX polyacrylamide gels (Bio-Rad, Hercules, CA, USA) with 5 µg of protein per lane, and electrophoresed at 200 V for 20 min. Three biological replicates per treatment were analyzed. Gels were stained with Bio-Safe™ Coomassie Stain (CBB, Bio-Rad, Hercules, CA, USA), and protein bands were quantified by densitometry using ImageJ (v.1.52, NIH, Bethesda, MD, USA) [106].

### 4.7. Immunodetection and Densitometric Analysis of Selected Proteins

Protein extracts, resolved as for CBB analyses, were transferred onto nitrocellulose membranes (0.45 μm or 0.2 μm pore size; Bio-Rad, Hercules, CA, USA) by semi-dry electroblotting (1.5 mA cm^−2^, 20 min). Membranes were air-dried, blocked for 1 h at room temperature (RT) with 5% non-fat dry milk (Bio-Rad, Hercules, CA, USA), and incubated for 1 h at RT with gentle agitation, followed by overnight incubation at 4 °C with primary antibodies specific to: RCA (AS10 700; 1:5000, 5 μg of protein per lane), PsbO (AS06 142-33, 1:5000, 5 μg of protein per lane—ppl), PsbA (D1, AS05 084; 1:1000, 5 μg ppl), Lhcb1 (AS01 004; 1:1000, 15 μg ppl), Lhcb5 (CP26, AS01 009; 1:1000, 5 μg ppl), PsaB (AS10 695; 1:1000, 5 μg ppl), Lhca4 (AS01 008; 1:1000, 15 μg ppl), SAG12 (AS14 2771; 1:1000, 25 μg of protein per lane), and eEF1a (AS10 934; 1:5000, 5 μg of protein per lane, loading control) (all from Agrisera, Vännäs, Sweden). After washing with TTBS buffer (0.05% Tween 20, 20 mM Tris, 500 mM NaCl), membranes were incubated for 1 h at RT with horseradish peroxidase-conjugated secondary antibody (AS09 602; 1:5000–1:10000), followed by colorimetric detection using the Pierce™ DAB Substrate Kit (Thermo Fisher Scientific, Waltham, MA, USA) for 5–10 min. Protein bands intensities were quantified via densitometry using ImageJ (v.1.52, NIH) [107]. For RCA, the combined signal from both α- and β-isoforms was analyzed as total RCA content (RCA_total_). Each treatment was analyzed in triplicate.

We analyzed the following proteins: ribulose-1,5-bisphosphate carboxylase/oxygenase activase (RCA), a chloroplast enzyme that activates Rubisco through ATP-dependent conformational changes; PsbO, an extrinsic subunit of PSII that stabilizes the water-splitting system; PsbA (D1 protein), a core PSII protein essential for charge separation and electron flow; Lhcb1, the major PSII light-harvesting protein abundant in thylakoid membranes; Lhcb5, a minor PSII antenna protein contributing to light capture and photoprotection; PsaB, a core PSI protein binding cofactors for electron transfer; Lhca4, a conserved PSI antenna protein facilitating energy transfer; SAG12, a senescence-specific cysteine protease and a well-established marker of leaf senescence; and eEF1α (elongation factor 1-alpha), a cytoplasmic translation factor commonly used as a stable reference.

### 4.8. Determination of Hydrogen Peroxide, Superoxide Dismutase Activity and Lipid Peroxidation Levels

The endogenous H_2_O_2_ content was determined following the previously described protocol [108]. Briefly, 150 mg of leaf tissue was immediately frozen in liquid nitrogen and ground to a fine powder. Samples were homogenized in 1 mL of extraction solution comprising 250 μL of 0.1% (*w*/*v*) trichloroacetic acid (TCA), 500 μL of 1 M KI, and 250 μL of 10 mM potassium phosphate buffer (pH 7.0) at 4 °C in darkness for 10 min. Homogenates were centrifuged at 12,000× *g* for 15 min at 4 °C, and 200 μL of the supernatant was incubated in darkness at room temperature for 20 min. Samples were subsequently diluted 10-fold with distilled water, and absorbance was measured at 350 nm. Quantification was based on a standard calibration curve prepared with H_2_O_2_ in 0.1% TCA. Results were expressed as μmol H_2_O_2_ per gram fresh weight (μmol g^−1^ FW). Each treatment was analyzed in six replicates.

SOD activity was determined spectrophotometrically using a modified method [109]. Leaf tissue (200 mg) was homogenized in cold 100 mM phosphate buffer (pH 7.8) containing 1 mM EDTA and 5% (*w*/*v*) PVPP. The homogenate was centrifuged at 10,000× *g* for 30 min at 4 °C, and the supernatant was used for analysis. Each assay mixture contained the enzyme extract, 0.75 mM NBT, 0.5 M EDTA (pH 8.0), and 0.1 mM riboflavin. After illumination (10 min, 40 µmol m^−2^ s^−1^), absorbance (A) was measured at 560 nm. One unit (U) of SOD activity was defined as the amount of enzyme required to inhibit 50% of NBT photoreduction. SOD activity was expressed in units (U g^−1^ FW), where one unit (1 U) is defined as the amount of enzyme required to inhibit NBT photoreduction by 50%, calculated as:$$1\;U=\left[\frac{\frac{A_{560}C-A_{560}S}{A_{560}C}}{50}\right]\times 100$$
where

A_X_—absorbance at wavelength x (nm);

C—control sample exposed to light;

S—test sample.

Lipid peroxidation, indicative of oxidative membrane damage, was assessed by measuring malondialdehyde (MDA) levels via thiobarbituric acid (TBA) reaction products (TBARS), following established methods [109]. For each replicate, 200 mg of leaf tissue was homogenized in 1 mL of methanol and incubated at 60 °C for 30 min. After centrifugation (10,000× *g*, 5 min), 300 μL of the supernatant was combined with 600 μL of a TCA-BHT-TBA mixture (0.18 M TCA, 65.5 μM BHT, and 45 mM TBA). The reaction mixtures were incubated at 95 °C for 5 min, then centrifuged again (10,000× *g*, 1 min). Absorbance of the supernatant was measured at 532 nm, with corrections for non-specific absorbance at 450 nm and 600 nm. MDA concentration (μmol g^−1^ FW) was calculated using the formula:$$MDA=6.45\times (A_{532}-A_{600})-0.56\times A_{450}$$

### 4.9. Statistical Analysis

All statistical analyses were conducted using Statistica 13.3 software (StatSoft Inc., Tulsa, OK, USA). Normality of data distribution was assessed with the Shapiro–Wilk test, while homogeneity of variances was verified using Levene’s test. Differences between experimental groups were evaluated by one-way analysis of variance (ANOVA), followed by Tukey’s HSD post hoc test. Results are expressed as means ± standard deviation (SD). Statistical significance was accepted at *p* < 0.05.

## 5. Conclusions

This study reveals distinct senescence dynamics among barley cultivars and their differential responsiveness to CK modulation. The winter cultivar Bursztyn was characterized as a previously unrecognized stay-green phenotype with delayed chlorophyll loss, sustained photochemical efficiency, and reduced oxidative damage, largely independent of exogenous CK supply. In contrast, Carina displayed rapid senescence but responded strongly to BA, which stabilized photosystems, enhanced antioxidant defenses, and mitigated respiratory decline. Lomerit exhibited intermediate traits, with partial responsiveness to BA. Functional and protein-level analyses demonstrated that photosynthetic decline was mainly linked to impaired Calvin cycle activity, while BA conferred cultivar-specific protection by stabilizing photosynthetic complexes and modulating energy partitioning. Inhibition of endogenous CK biosynthesis further highlighted differential reliance on biosynthetic pathways across cultivars.

Overall, these findings establish Bursztyn as a valuable genetic resource for dissecting delayed senescence and emphasize that CK-mediated enhancement of stress resilience operates in a genotype-dependent manner. They underscore the potential of tailored CK-based strategies to prolong photosynthetic competence and improve cereal stress tolerance.

## Figures and Tables

**Figure 1 ijms-26-09749-f001:**
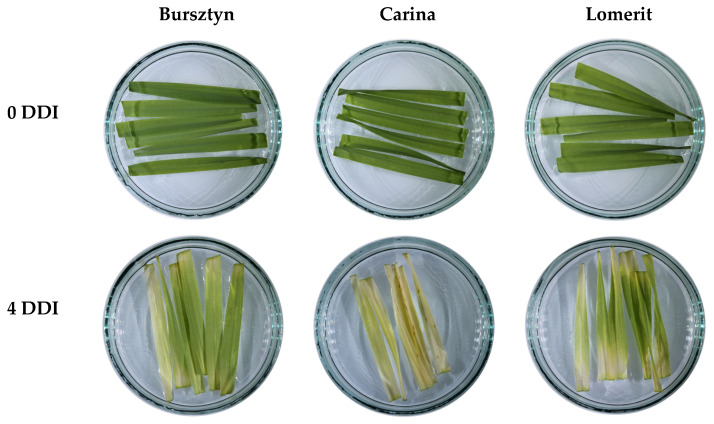
Images showing 50 mm segments of the first true leaf of *Hordeum vulgare* L. cultivars: Bursztyn (winter cv.), Carina (spring cv.) and Lomerit (winter cv.), taken immediately prior to incubation—0 DDI (days of dark incubation) and after 96 h of dark incubation—4 DDI in 0.2% DMSO (DIS). Leaf segments were incubated in 85 mm diameter Petri dishes containing 15 cm^3^ of deionized water supplemented with DMSO. Images were captured using a Canon EOS 1300D camera (Canon Inc., Tokyo, Japan).

**Figure 2 ijms-26-09749-f002:**
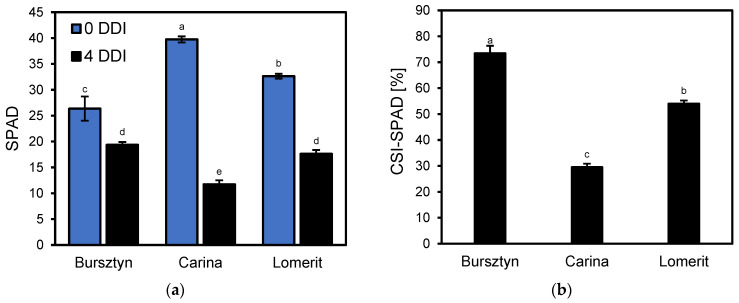
(**a**) Relative chlorophyll content (SPAD) and (**b**) relative chlorophyll stability (CSI-SPAD), estimated based on the ratio of the SPAD leaf greenness index of the first true leaf of *H. vulgare* L., cultivars: Bursztyn, Carina and Lomerit measured in 50 mm segments directly before incubation (0 DDI, blue bars) and after 4 days (96 h) of incubation in darkness (4 DDI, black bars). Leaf segments were incubated in 0.2% (*v*/*v*) DMSO. Bars represent means ± standard deviations for *n* = 20. Different letters indicate statistically significant differences between means according to Tukey’s HSD test (*p* < 0.05).

**Figure 3 ijms-26-09749-f003:**
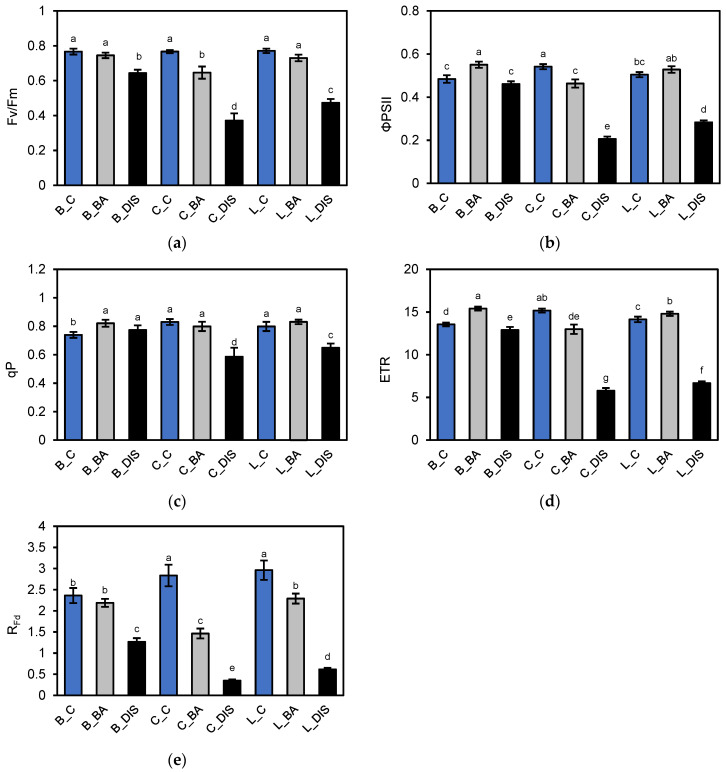
Evaluation of photosynthetic activity of photosystem II (PSII) in 50 mm segments of the first true leaf of *H. vulgare* L. cultivars Bursztyn (B), Carina (C), and Lomerit (L) after 72 h of incubation under light in 0.2% DMSO (control, C, blue bars), or in darkness in a solution of 50 µM BA in 0.2% DMSO (BA, gray bars) and in 0.2% DMSO (DIS, black bars). (**a**) Fv/Fm: maximum quantum efficiency of PSII photochemistry, determined after dark adaptation; (**b**) ΦPSII: effective quantum yield of PSII photochemistry in illuminated leaves; (**c**) qP: photochemical quenching coefficient according to the puddle model; (**d**) ETR: electron transport rate through PSII; (**e**) R_Fd_: chlorophyll fluorescence decrease ratio. Bars represent means ± standard deviations for *n* = 12. Different letters indicate statistically significant differences between means, determined using Tukey’s HSD test (*p* < 0.05).

**Figure 4 ijms-26-09749-f004:**
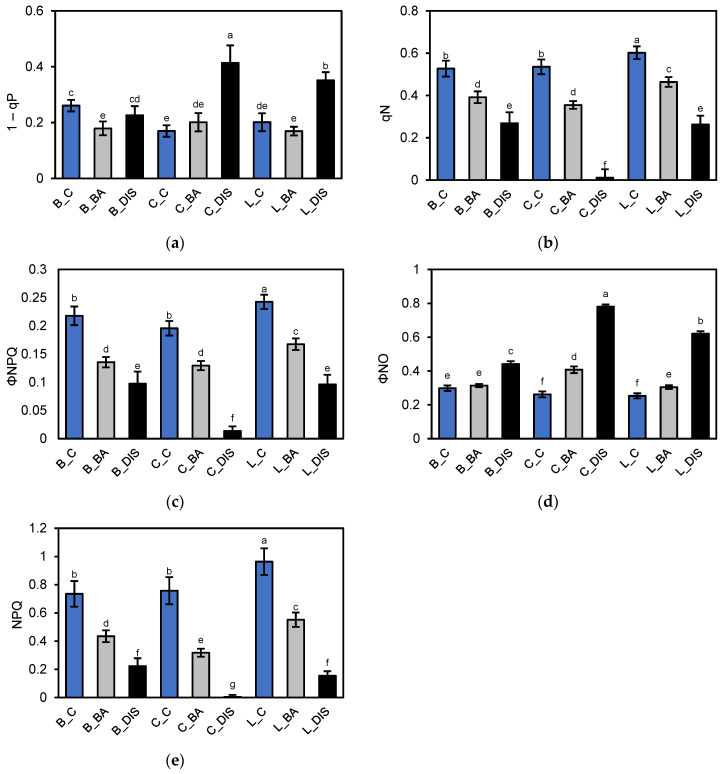
Evaluation of photosynthetic activity of photosystem II (PSII) in 50 mm segments of the first true leaf of *H. vulgare* L. cultivars Bursztyn (B), Carina (C), and Lomerit (L) after 72 h of incubation under light in 0.2% DMSO (control, C, blue bars), or in darkness in a solution of 50 µM BA in 0.2% DMSO (BA, gray bars) and in 0.2% DMSO (DIS, black bars). (**a**) 1–qP: PSII excitation pressure based on the qP coefficient; (**b**) qN: non-photochemical quenching coefficient in PSII; (**c**) ΦNPQ: quantum yield of regulated non-photochemical energy dissipation; (**d**) ΦNO: quantum yield of non-regulated non-photochemical energy dissipation; (**e**) NPQ: non-photochemical quenching in PSII. Bars represent means ± standard deviations for *n* = 12. Different letters indicate statistically significant differences between means, determined using Tukey’s HSD test (*p* < 0.05).

**Figure 5 ijms-26-09749-f005:**
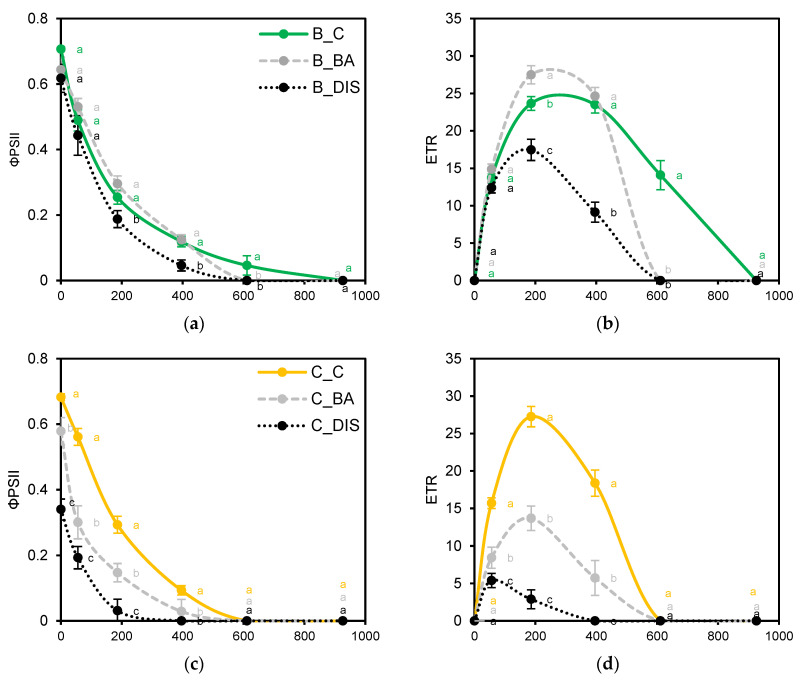
Evaluation of photosystem II (PSII) photosynthetic activity in response to increasing photosynthetic photon flux density (PPFD, 0–926 µmol m^−2^ s^−1^) in 50 mm segments of the first true leaf of *H. vulgare* L. cultivars Bursztyn (B, (**a,b**), Carina (C, (**c**,**d**)), and Lomerit (L, (**e**,**f**)) after 72 h of incubation under light in 0.2% DMSO (control, C, Bursztyn: green solid line, Carina: yellow solid line, Lomerit: red solid line), or in darkness in a solution of 50 µM BA in 0.2% DMSO (BA, dashed gray line) and in 0.2% DMSO (DIS, dotted black line). (**a**,**c**,**e**) ΦPSII: effective quantum yield of PSII photochemistry in illuminated leaves; (**b**,**d**,**f**) ETR: electron transport rate through PSII. Points represent means ± standard deviations for *n* = 12. Different letters indicate statistically significant differences between means of the same light intensity, determined using Tukey’s HSD test (*p* < 0.05).

**Figure 6 ijms-26-09749-f006:**
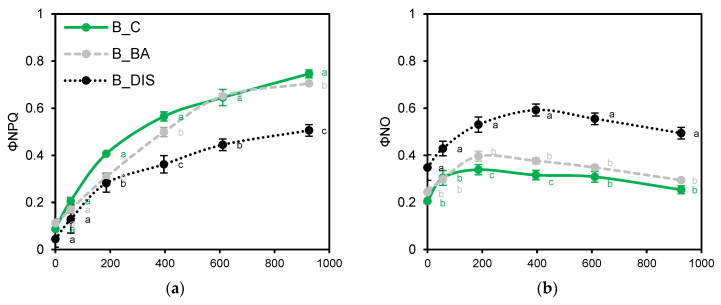
Evaluation of photosystem II (PSII) photosynthetic activity in response to increasing photosynthetic photon flux density (PPFD, 0–926 µmol m^−2^ s^−1^) in 50 mm segments of the first true leaf of *H. vulgare* L. cultivars Bursztyn (B, (**a**,**b**)), Carina (C, (**c**,**d**)), and Lomerit (L, (**e**,**f**)) after 72 h of incubation under light in 0.2% DMSO (control, C, Bursztyn: green solid line, Carina: yellow solid line, Lomerit: red solid line), or in darkness in a solution of 50 µM BA in 0.2% DMSO (BA, dashed gray line) and in 0.2% DMSO (DIS, dotted black line). (**a**,**c**,**e**) ΦNPQ: quantum yield of regulated non-photochemical energy dissipation; (**b**,**d**,**f**) ΦNO: quantum yield of non-regulated non-photochemical energy dissipation. Points represent means ± standard deviations for *n* = 12. Different letters indicate statistically significant differences between means of the same light intensity, determined using Tukey’s HSD test (*p* < 0.05).

**Figure 7 ijms-26-09749-f007:**
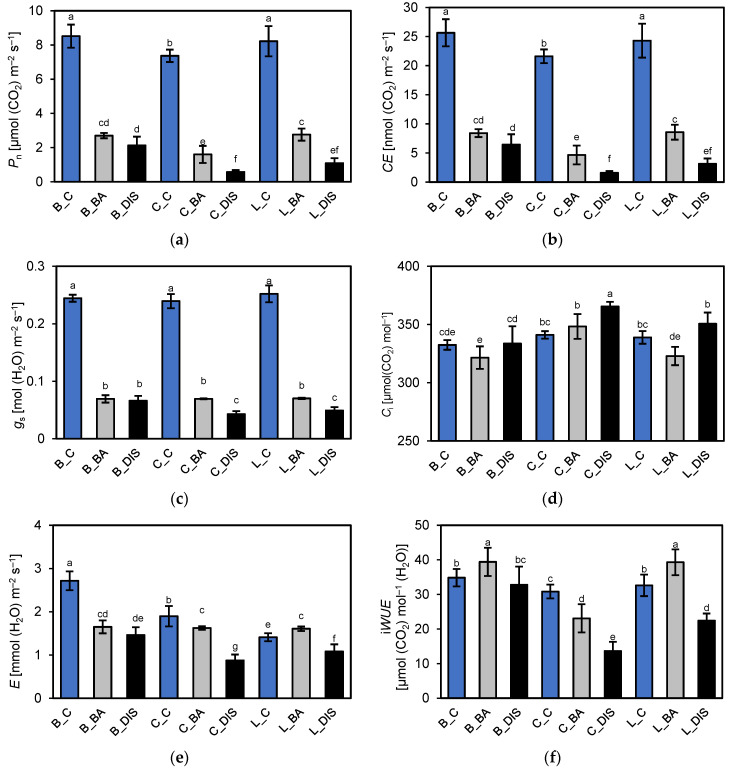
Instantaneous gas exchange parameters assessed directly under growth conditions in the first true leaf of *H. vulgare* L. cultivars Bursztyn (B), Carina (C), and Lomerit (L) after 96 h of incubation under light, preceded by two foliar applications of distilled water (H_2_O_d_) with 0.2% DMSO (control, C, blue bars), or in darkness, preceded by two foliar applications of 50 µM BA in 0.2% DMSO (BA, gray bars), as well as treatments preceded by two applications of H_2_O_d_ with 0.2% DMSO (DIS, black bars). (**a**) *P*_n_: net CO_2_ assimilation rate; (**b**) *CE*: carboxylation efficiency; (**c**) *g*_s_: stomatal conductance; (**d**) *C*_i_: intercellular CO_2_ concentration; (**e**) *E*: transpiration rate; (**f**) i*WUE*: intrinsic water use efficiency (*P*_n_/*g*_s_). Bars represent means ± standard deviations for *n* = 12. Different letters indicate statistically significant differences between means, determined using Tukey’s HSD test (*p* < 0.05).

**Figure 8 ijms-26-09749-f008:**
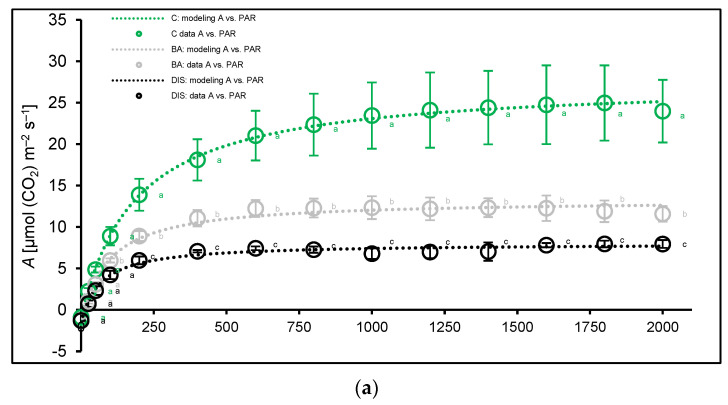
Net CO_2_ assimilation rate (*A*) in response to increasing PPFD in the first true leaf of *H. vulgare* L. cultivars (**a**) Bursztyn, (**b**) Carina, and (**c**) Lomerit after 96 h of light incubation preceded by two foliar sprays with H_2_O_d_ containing 0.2% DMSO (control, C, Bursztyn: green dotted line, Carina: yellow dotted line, Lomerit: red dotted line), or following dark incubation preceded by two foliar applications of 50 µM BA in 0.2% DMSO (BA, gray dotted line), and two sprays with H_2_O_d_ containing 0.2% DMSO (DIS, black dotted line). Circles represent means ± standard deviations for *n* = 6, along with the nonlinear regression model (dotted line) fitted to the empirical data of CO_2_ assimilation (*A*) versus PAR. Different letters indicate statistically significant differences between means, determined using Tukey’s HSD test (*p* < 0.05).

**Figure 9 ijms-26-09749-f009:**
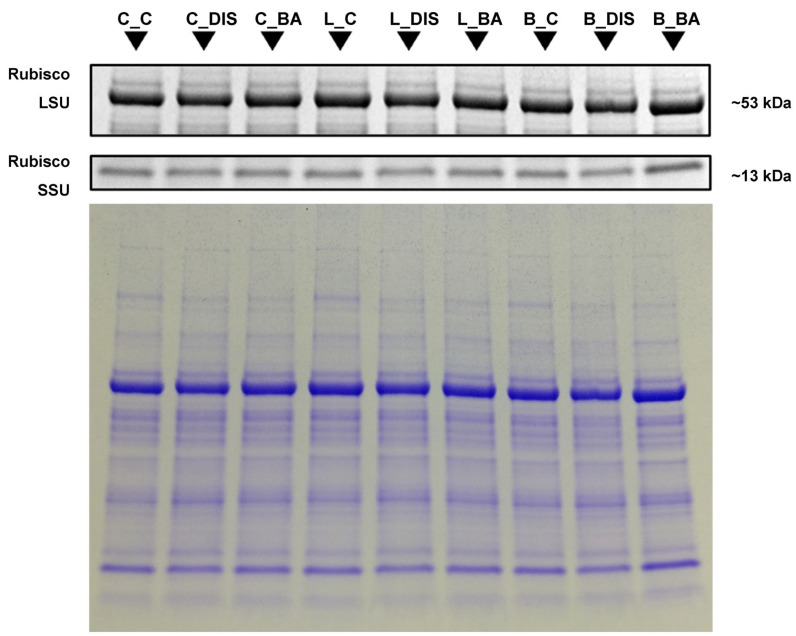
Coomassie Brilliant Blue (CBB) staining analysis of polyacrylamide gel for the identification of large (LSU) and small (SSU) subunits of Rubisco (top), following protein extraction from leaf powder obtained from 50 mm fragments of the first true leaf of *H. vulgare* L. cultivars Carina (C), Lomerit (L), and Bursztyn (B), after 72 h of incubation in light with 0.2% DMSO (control, C), or in darkness with 0.2% DMSO (DIS), and in a solution of 50 µM BA in 0.2% DMSO (BA). Protein samples (5 µg per well) were separated on a 4–20% TGX gradient polyacrylamide gel and visualized using CBB staining (bottom).

**Figure 10 ijms-26-09749-f010:**
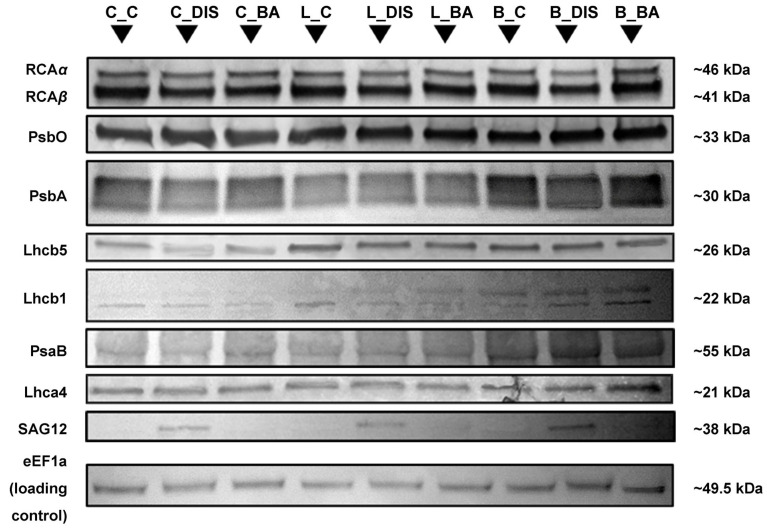
Western blot analysis of RCA protein (*α* and *β* isoforms), PSII-related proteins (PsbO, PsbA, Lhcb5, Lhcb1), PSI-related proteins (PsaB and Lhca4), the senescence-associated marker protein SAG12, and the loading control (eEF1a). Proteins were extracted from leaf powder obtained from 50 mm fragments of the first true leaf of *H. vulgare* L. cultivars Carina (C), Lomerit (L), and Bursztyn (B) after 72 h of incubation in light with 0.2% DMSO (control, C), in darkness with 0.2% DMSO (DIS), or in a solution of 50 µM BA in 0.2% DMSO (BA). For RCA, the combined signal from both *α*- and *β*-isoforms was analyzed as total RCA content (RCA_total_). Proteins were separated on a 4–20% TGX gradient polyacrylamide gel, transferred to a nitrocellulose membrane, and visualized using DAB staining.

**Figure 11 ijms-26-09749-f011:**
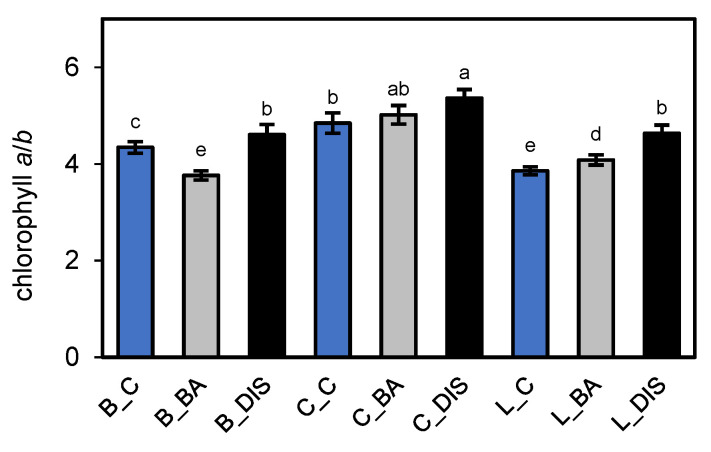
Chlorophyll *a* to *b* ratio in 50 mm fragments of the first true leaf of *H. vulgare* L. cultivars Bursztyn (B), Carina (C), and Lomerit (L) after 72 h of incubation under light in 0.2% DMSO (C, control, blue bars), or in darkness in the presence of 50 µM BA in 0.2% DMSO (BA, gray bars), 0.2% DMSO alone (DIS, black bars). Bars represent means ± standard deviations for *n* = 10. Different letters indicate statistically significant differences between means according to Tukey’s HSD test (*p* < 0.05).

**Figure 12 ijms-26-09749-f012:**
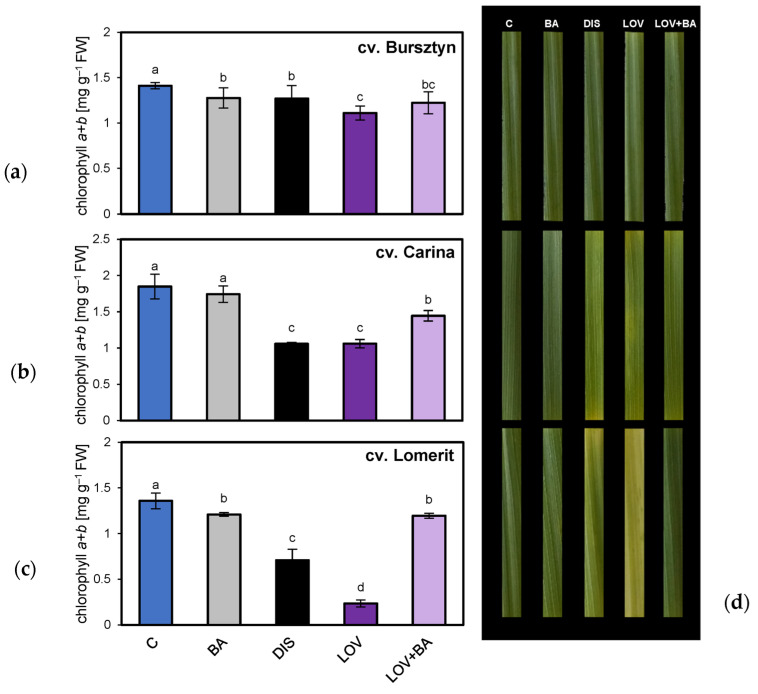
Chlorophyll *a* and *b* content (mg g^−1^ FW) in 50 mm fragments of the first true leaf of *H. vulgare* L. cultivars (**a**) Bursztyn, (**b**) Carina, and (**c**) Lomerit, and (**d**) the morphology of 50 mm fragments of the first true leaf of the respective cultivars after 72 h of incubation under light in 0.2% DMSO (control, C, blue bars), or in darkness in the presence of 50 µM BA in 0.2% DMSO (BA, gray bars), 0.2% DMSO alone (DIS, black bars), 40 µM lovastatin in 0.2% DMSO (LOV, purple bars), or a combination of 40 µM lovastatin and 50 µM BA in 0.2% DMSO (LOV + BA, light purple bars). Bars (**a**–**c**) represent means ± standard deviations for *n* = 10. Different letters indicate statistically significant differences between means according to Tukey’s HSD test (*p* < 0.05). FW—fresh weight. Images of 50 mm fragments of the first true leaf of *H. vulgare* L. cultivars (**d**) were taken after 72 h incubation using a Canon EOS 1300D camera (Canon Inc., Tokyo, Japan).

**Figure 13 ijms-26-09749-f013:**
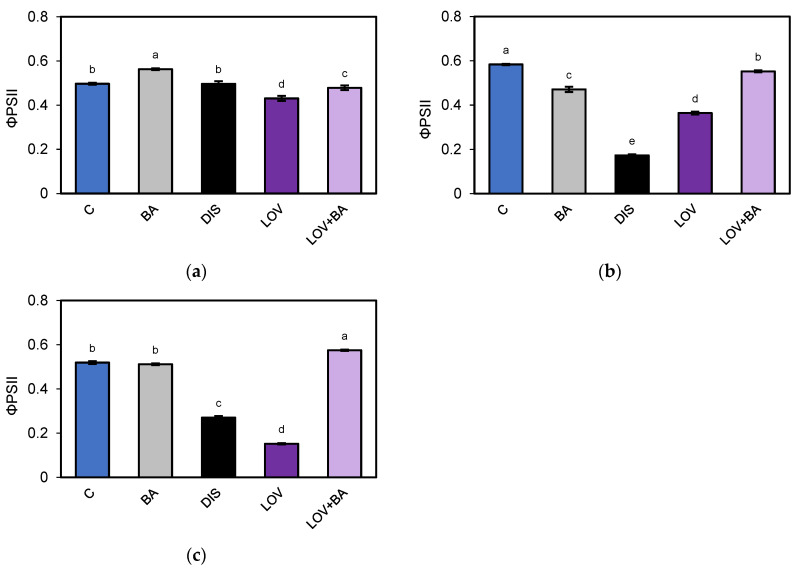
Evaluation of ΦPSII: effective quantum yield of PSII photochemistry in illuminated leaves in 50 mm segments of the first true leaf of *H. vulgare* L. cultivars (**a**) Bursztyn, (**b**) Carina, and (**c**) Lomerit after 72 h of incubation under light in 0.2% DMSO (control, C, blue bars), or in darkness in the presence of 50 µM BA in 0.2% DMSO (BA, gray bars), 0.2% DMSO alone (DIS, black bars), 40 µM lovastatin in 0.2% DMSO (LOV, purple bars), or a combination of 40 µM lovastatin and 50 µM BA in 0.2% DMSO (LOV + BA, light purple bars). Bars represent means ± standard deviations for *n* = 12. Different letters indicate statistically significant differences between means, determined using Tukey’s HSD test (*p* < 0.05).

**Figure 14 ijms-26-09749-f014:**
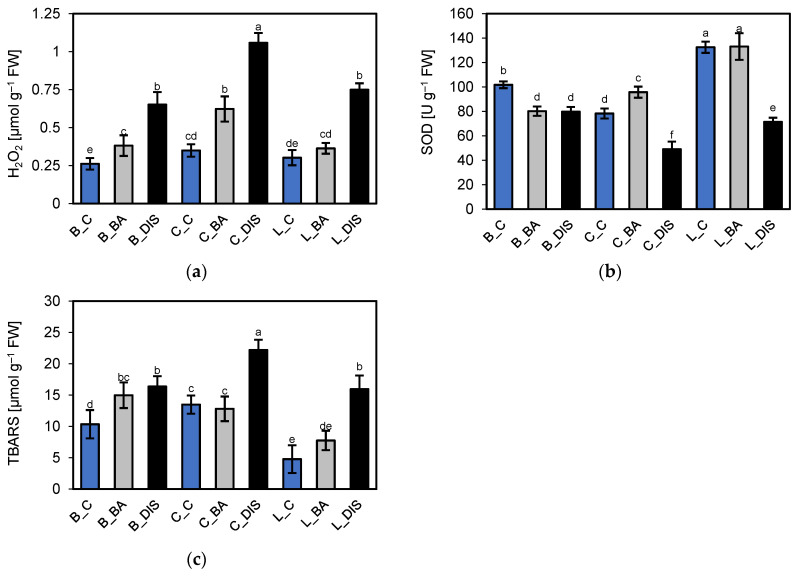
Effect of 72 h incubation of 50 mm segments of the first true leaf of *H. vulgare* L. cultivars Bursztyn (B), Carina (C), and Lomerit (L) under light in 0.2% DMSO (C, control, blue bars), or in darkness in the presence of 50 µM BA in 0.2% DMSO (BA, gray bars), 0.2% DMSO alone (DIS, black bars) on (**a**) endogenous hydrogen peroxide (H_2_O_2_) content; (**b**) superoxide dismutase (SOD) activity; and (**c**) membrane lipid peroxidation intensity (TBARS). Bars represent means ± standard deviations for *n* = 6. Different letters indicate statistically significant differences between means according to Tukey’s HSD test (*p* < 0.05). FW—fresh weight.

**Figure 15 ijms-26-09749-f015:**
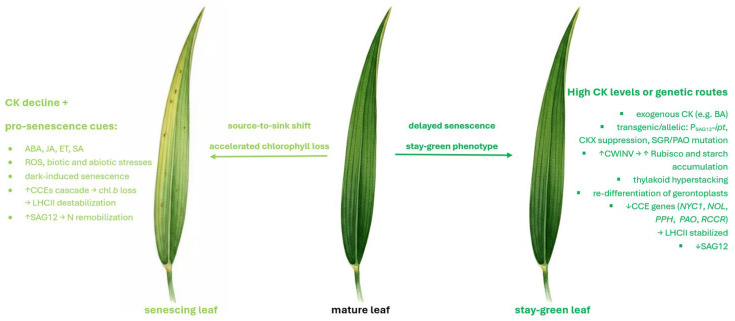
A mature barley leaf may follow two alternative trajectories. A decline in cytokinin (CK) levels, together with pro-senescence signals such as abscisic acid (ABA), jasmonic acid (JA), ethylene (ET) and salicylic acid (SA), ROS, environmental stresses or DIS, activates the chlorophyll catabolic pathway, leading to Chl *b* degradation, destabilization of LHCII, pigment loss, activation of proteases (e.g., SAG12) and, ultimately, nitrogen remobilization during leaf senescence. In contrast, elevated CK levels or specific genetic modifications (e.g., P_SAG12_-*ipt*, CKX suppression, *SGR*/*PAO* mutations) delay senescence by stimulating CWINV activity (increasing Rubisco content and starch accumulation), promoting thylakoid hyperstacking and inducing gerontoplast re-differentiation. CKs also repress chlorophyll catabolic genes (*NYC1*, *NOL*, *PPH*, *PAO*, *RCCR*), stabilize LHCII-Chl *b* complexes and inhibit protease activation. Collectively, these processes maintain chlorophyll levels and confer a stay-green phenotype.

**Table 1 ijms-26-09749-t001:** Parameters of the nonlinear regression model estimated from the fitting of empirical gas exchange data (*A*/PAR relationship) in the first true leaf of *H. vulgare* L. cultivars Bursztyn (B), Carina (C), and Lomerit (L) after 96 h of incubation in light following two foliar applications of H_2_O_d_ with 0.2% DMSO (control, C), or after 96 h of incubation in darkness following two foliar applications of 50 µM benzyladenine (BA) in 0.2% DMSO, and H_2_O_d_ with 0.2% DMSO (DIS). *P*_gmax_: gross photosynthesis; *P*_nmax_: maximal net photosynthesis; *R*_d_: dark respiration; Φ: quantum yield of CO_2_ assimilation; *I*_0_: 0 µmol m^−2^ s^−1^ PPFD, *I*_comp_: light compensation point; *I*_200_: 200 µmol m^−2^ s^−1^ PPFD.

Parameter	Cultivar and Treatment								
	B_C	B_BA	B_DIS	C_C	C_BA	C_DIS	L_C	L_BA	L_DIS
*P*_gmax_ [µmol(CO_2_) m^−2^ s^−1^]	28.63 ± 3.16 ^b^	15.04 ± 1.87 ^c^	9.49 ± 1.11 ^d^	29.88 ± 3.34 ^b^	7.14 ± 0.68 ^e^	2.00 ± 0.32 ^g^	35.33 ± 3.95 ^a^	10.70 ± 1.14 ^d^	5.78 ± 0.84 ^f^
*P*_nmax_ [µmol(CO_2_) m^−2^ s^−1^]	25.07 ± 2.43 ^b^	12.60 ± 1.49 ^c^	7.68 ± 0.89 ^d^	25.52 ± 2.62 ^b^	5.12 ± 0.54 ^e^	1.46 ± 0.13 ^f^	30.50 ± 3.86 ^a^	8.68 ± 0.87 ^d^	4.41 ± 0.59 ^e^
*R*_d_ [µmol(CO_2_) m^−2^ s^−1^]	1.08 ± 0.14 ^c^	1.77 ± 0.11 ^a^	1.49 ± 0.24 ^b^	1.52 ± 0.12 ^b^	1.78 ± 0.21 ^a^	0.44 ± 0.12 ^d^	1.72 ± 0.23 ^a^	1.65 ± 0.27 ^ab^	1.22 ± 0.12 ^c^
*I*_comp_ [µmol(photons) m^−2^ s^−1^]	7.25 ± 0.76 ^e^	12.03 ± 1.43 ^c^	12.63 ± 1.47 ^c^	10.99 ± 1.24 ^d^	23.16 ± 2.56 ^b^	29.62 ± 3.65 ^a^	9.65 ± 1.17 ^d^	12.44 ± 1.45 ^c^	13.40 ± 1.28 ^c^
Φ*_I_*_0–*I*comp_ [µmol(CO_2_) µmol(photons)^–1^]	0.149 ± 0.02 ^b^	0.147 ± 0.03 ^b^	0.119 ± 0.02 ^c^	0.139 ± 0.024 ^b^	0.076 ± 0.013 ^d^	0.015 ± 0.004 ^e^	0.179 ±0.02 ^a^	0.133 ±0.02 ^b^	0.091 ± 0.011 ^d^
Φ*_I_*_comp–*I*200_ [µmol(CO_2_) µmol(photons)^–1^]	0.068 ± 0.01 ^a^	0.041 ± 0.01 ^b^	0.026 ± 0.004 ^c^	0.067 ±0.013 ^a^	0.018 ± 0.004 ^d^	0.005 ± 0.001 ^e^	0.083 ± 0.012 ^a^	0.029 ± 0.01 ^c^	0.015 ± 0.003 ^d^

The presented values are means of six replicates ± SD. Different letters (a–g) in the same row indicate significant differences between treatments at *p* < 0.05 with Tukey’s HSD test.

**Table 2 ijms-26-09749-t002:** Relative protein content (AU) estimated by densitometric analysis of bands visualized on gel/membrane following electrophoretic separation of proteins isolated from leaf powder of 50 mm fragments of the first true leaf of *H. vulgare* L. cultivars Carina (C), Lomerit (L), and Bursztyn (B) after 72 h incubation under light in 0.2% DMSO (control), in darkness in 0.2% DMSO (DIS), or in 50 µM BA solution in 0.2% DMSO (BA).

Protein	Relative Abundance [AU] ± SD								
	C_C	C_DIS	C_BA	L_C	L_DIS	L_BA	B_C	B_DIS	B_BA
Rubisco LSU	12.07 ± 0.30 ^c^	11.72 ± 0.22 ^d^	12.58 ± 0.14 ^b^	13.19 ± 0.55 ^a^	12.16 ± 0.15 ^c^	12.79 ± 0.41 ^b^	12.09 ± 0.17 ^c^	11.67 ± 0.41 ^d^	12.37 ± 0.30 ^b^
Rubisco SSU	3.00 ± 0.04 ^b^	2.67 ± 0.03 ^d^	2.82 ± 0.06 ^c^	2.92 ± 0.12 ^b^	2.76 ± 0.13 ^cd^	2.97 ± 0.08 ^b^	2.87 ± 0.13 ^bc^	2.64 ± 0.03 ^d^	3.62 ± 0.10 ^a^
RCA_total_	11.19 ± 0.20 ^bc^	8.75 ± 0.13 ^f^	10.57 ± 0.22 ^d^	11.66 ± 0.29 ^b^	10.05 ± 0.36 ^e^	10.44 ± 0.29 ^d^	10.95 ± 0.17 ^c^	8.10 ± 0.37 ^g^	12.63 ± 0.23 ^a^
PsbO	8.88 ± 0.39 ^b^	8.85 ± 0.34 ^b^	8.10 ± 0.20 ^cd^	9.73 ± 0.14 ^a^	8.43 ± 0.34 ^bc^	7.99 ± 0.27 ^d^	7.63 ± 0.30 ^e^	7.52 ± 0.09 ^e^	7.87 ± 0.20 ^d^
PsbA	9.24 ± 0.09 ^a^	6.98 ± 0.08 ^d^	8.63 ± 0.37 ^b^	5.25 ± 0.14 ^g^	5.59 ± 0.11 ^f^	5.61 ± 0.23 ^f^	7.70 ± 0.35 ^c^	6.30 ± 0.14 ^e^	7.63 ± 0.09 ^c^
Lhcb5	2.33 ± 0.11 ^b^	1.47 ± 0.04 ^e^	1.78 ± 0.05 ^d^	2.81 ± 0.08 ^a^	1.96 ± 0.09 ^c^	1.97 ± 0.05 ^c^	1.86 ± 0.03 ^c^	1.91 ± 0.08 ^c^	1.68 ± 0.05 ^d^
Lhcb1	14.26 ± 0.52 ^b^	5.75 ± 0.25 ^f^	7.44 ± 0.32 ^e^	13.89 ± 0.27 ^b^	11.76 ± 0.41 ^d^	12.74 ± 0.22 ^c^	12.44 ± 0.62 ^c^	11.76 ± 0.48 ^d^	15.58 ± 0.58 ^a^
PsaB	5.64 ± 0.06 ^b^	4.19 ± 0.06 ^e^	6.01 ± 0.23 ^a^	4.75 ± 0.06 ^d^	3.70 ± 0.04 ^f^	3.71 ± 0.11 ^f^	5.10 ± 0.25 ^c^	4.61 ± 0.06 ^d^	5.52 ± 0.19 ^b^
Lhca4	2.28 ± 0.03 ^d^	2.07 ± 0.06 ^e^	2.56 ± 0.09 ^b^	2.12 ± 0.10 ^e^	2.17 ± 0.07 ^e^	2.44 ± 0.11 ^bc^	2.36 ± 0.12 ^cd^	2.23 ± 0.05 ^d^	2.85 ± 0.09 ^a^
SAG12	0.00 ± 0.00 ^d^	18.91 ± 0.35 ^a^	0.00 ± 0.00 ^d^	0.00 ± 0.00 ^d^	16.64 ± 0.34 ^b^	0.00 ± 0.00 ^d^	0.00 ± 0.00 ^d^	15.65 ± 0.51 ^c^	0.00 ± 0.00 ^d^
eEF1a	3.73 ± 0.15 ^a^	3.86 ± 0.12 ^a^	3.88 ± 0.15^a^	3.05 ± 0.06 ^bc^	2.82 ± 0.11 ^c^	2.94 ± 0.06 ^c^	2.96 ± 0.03 ^c^	3.16 ± 0.07 ^b^	3.01 ± 0.13 ^bc^

The presented values are means of three replicates ± SD. Different letters (a–g) in the same row indicate significant differences between treatments at *p* ≤ 0.05 with Tukey’s HSD test.

**Table 3 ijms-26-09749-t003:** Chlorophyll *a* fluorescence parameters in PSII estimated using the Maxi IMAGING PAM M Series system.

Parameter	Equation	Definition	Reference
**Fv/Fm**	Fv/Fm = (Fm − Fo)/Fm	maximum quantum efficiency of PSII photochemistry after dark adaptation	[96]
**ΦPSII**	ΦPSII = (Fm′ − F)/Fm′	effective quantum yield of PSII in light-adapted state *	[97]
**qP**	qP = (Fm′ − F)/(Fm′ − Fo′)	photochemical quenching coefficient of PSII based on the puddle model	[98]
**ETR**	ETR = ΦPSII × PAR × Abs × 0.5	electron transport rate through PSII	[99]
**R_Fd_**	R_Fd_ = (Fm – Fs **)/Fs	fluorescence decrease ratio (vitality index)	[94]
**ΦNPQ**	ΦNPQ = 1 − ΦPSII − 1/[NPQ + 1 + qL (Fm/Fo − 1)]	quantum yield of regulated non-photochemical energy dissipation	[98]
**ΦNO**	ΦNO = 1/[NPQ + 1 + qL (Fm/Fo − 1)]	quantum yield of non-regulated (constitutive) energy dissipation	[98]
**NPQ**	NPQ = (Fm − Fm′)/Fm′	non-photochemical quenching in PSII	[100]
**qN**	qN = (Fm − Fm′)/(Fm − Fo′)	coefficient of non-photochemical quenching in PSII	[31]
**1 − qP**	1 − qP = 1 − [(Fm′ − F)/(Fm′ − Fo′)]	PSII excitation pressure based on photochemical quenching	[99]

* where ΦPSII + ΦNPQ + ΦNO = 1. ** the parameter Fs, in the case of the Maxi IMAGING PAM M-Series fluorometer, is denoted as F [101].

## Data Availability

The data presented in this study are available on request from the corresponding author. The data are not publicly available due to the strict management of various data and technical resources within the research teams.

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
