# Peer review of "Mechanistic Insights into Cytokinin-Regulated Leaf Senescence in Barley: Genotype-Specific Responses in Physiology and Protein Stability"

_ijms, 2025, doi:10.3390/ijms26199749_

Round 1
Reviewer 1 Report
Comments and Suggestions for Authors
In the manuscript named “Mechanistic Insights into Cytokinin-Regulated Leaf Senescence in Barley: Genotype-Specific Responses in Physiology, Photosynthesis, and Protein Stability”, authors had performed comprehensive experiments to systematically investigate the physiological, biochemical, and protein-level changes during leaf development in three barley cultivars. They had received favorable experimental results, and their results had highlighted distinct regulatory networks shaping CK-mediated senescence responses in barely. These findings held significant suggestions for barley and other crops genetic improvement in future, but there were comments about it before publish.
(1) The authors had demonstrated the differences in leaf development among three barley cultivars at multiple levels, including physiological, biochemical, and protein levels. However, the intermediate steps, such as the mRNA of functional genes, were not presented. It was unclear whether the authors had also completed research about it, it was recommended to supply these evidences if they were accomplished.
(2) The authors had primarily focused on CK (cytokinins) regulation pathways in leaf development, but authors had well displayed relative phenomenon, for example, photosynthetic pigments, ROS, etc. However, the regulatory mechanisms underlying this process had not well addressed. It was recommended that the authors reviewed keys genes in these processes, and supplied their expressions or other evidences to improve readers knowledge. For example, how did CKs downregulate CCEs, etc.
(3) Based on the CSI-SPAD results, the spring cv had exhibited a more intense response. It remained unclear whether cytokinins regulate light- or temperature-related pathways, thereby influencing chlorophyll metabolism. It was recommended that the authors could supplement relevant research to address this question.
(4) The conclusion section of the manuscript was overly redundant. It was recommended that the authors would strengthen the refinement of summary statements and reduce the length by at least half.
(5) Figure 2 lacked a legend, it was recommended to add a legend. Other figures had similar issues, please check them. In addition, some figures had present in two print pages, please adjust them.
(6) In figure 10, each protein had indexed with gene names, the subplot names, for example, a, b, etc, could be deleted, all western blot gels could be merged into one figure, not sub-figures.
(7) It was recommended to add a mechanism figure to summary CK function in leaf development.
(8) “distilled water (H2Od)” (line 841)? Please check it.
Author Response
Please see the attachment.
Manuscript ID: ijms-3872028
- Reviewer 1
All changes in the manuscript have been highlighted in yellow for clarity.
Comments 1: (1) The authors had demonstrated the differences in leaf development among three barley cultivars at multiple levels, including physiological, biochemical, and protein levels. However, the intermediate steps, such as the mRNA of functional genes, were not presented. It was unclear whether the authors had also completed research about it, it was recommended to supply these evidences if they were accomplished;
Response 1: The analyses of mRNA expression patterns of functional genes during senescence have also been accomplished. However, as these analyses were performed in collaboration with other co-authors and include a substantial amount of additional data, they will be published in a separate manuscript together with other complementary analyses.
Comments 2: (2) The authors had primarily focused on CK (cytokinins) regulation pathways in leaf development, but authors had well displayed relative phenomenon, for example, photosynthetic pigments, ROS, etc. However, the regulatory mechanisms underlying this process had not well addressed. It was recommended that the authors reviewed keys genes in these processes, and supplied their expressions or other evidences to improve readers knowledge. For example, how did CKs downregulate CCEs, etc.
Response 2: Thank you for this valuable comment. Key information about CCEs has been added (highlighted in yellow, starting at line 600). Expression patterns of CCEs, as well as CK metabolism enzymes (CKX and IPT), have already been analyzed, but these results are included in another manuscript (as noted in Response 1). Including both protein- and gene-expression analyses in a single manuscript would considerably extend its length and reduce clarity. Nevertheless, the Reviewer’s suggestion is very important, especially since such gene-expression data in barley may be compared to previously reported results on rice leaf senescence during DIS, as presented in an earlier study (https://doi.org/10.1093/jxb/erv575.
Comments 3: (3) Based on the CSI-SPAD results, the spring cv had exhibited a more intense response. It remained unclear whether cytokinins regulate light- or temperature-related pathways, thereby influencing chlorophyll metabolism. It was recommended that the authors could supplement relevant research to address this question.
Response 3: Thank you for this comment. The three barley cultivars analyzed here were part of a broader screening program of spring and winter barley cultivars aimed at studying their senescence under field conditions. However, in the current manuscript, all cultivars were grown under stable, controlled conditions; therefore, temperature-related patterns of senescence measured with SPAD, as well as their connection to CKs, are not relevant to this dataset. Additional research related to field experiments, including temperature–CK interactions in barley cultivars, will be presented in another manuscript..
Comments 4: (4) The conclusion section of the manuscript was overly redundant. It was recommended that the authors would strengthen the refinement of summary statements and reduce the length by at least half.
Response 4: We thank the Reviewer for this valuable comment. Following the recommendation, the Conclusion section (line 1084) has been revised to remove redundancies, and its length has been reduced by approximately half. The revised version provides a concise synthesis of the main findings, emphasizing cultivar-specific senescence patterns and cytokinin responses, while avoiding unnecessary repetition.
Comments 5: (5) Figure 2 lacked a legend, it was recommended to add a legend. Other figures had similar issues, please check them. In addition, some figures had present in two print pages, please adjust them.
Response 5: Thank you for this comment. In Figure 2b (line 126), different colors were originally used to represent cultivars; however, since this was not explained in the figure legend, it could be misleading. We therefore changed Figure 2b to use black bars for CSI-SPAD values measured after 4 days (96 h) of dark incubation (4 DDI). Additional clarifying information has been included in the figure legends, and formatting issues with figures extending across two pages have been corrected where possible.
Comments 6: (6) In figure 10, each protein had indexed with gene names, the subplot names, for example, a, b, etc, could be deleted, all western blot gels could be merged into one figure, not sub-figures.
Response 6: Thank you for this comment. Figure 10 has been revised accordingly: subplot labels have been removed, and all western blot gels have been merged into a single figure (line 381).
Comments 7: (7) It was recommended to add a mechanism figure to summary CK function in leaf development.
Response 7: Thank you for this comment. A new figure (Figure 15, Discussion section, line 560) has been added to summarize CK functions specifically related to CK-mediated leaf senescence (line 545)
Comments 8: (8) “distilled water (H2Od)” (line 841)? Please check it.
Response 8: Thank you for this comment. The relevant section now reads:
“Samples were placed adaxial side up in 85 mm Petri dishes containing 15 mL of distilled water (H₂Od) with 0.2% DMSO (Honeywell International Inc., Charlotte, USA), supplemented with either 50 µM benzyladenine (BA) (Sigma-Aldrich, St. Louis, USA) or 40 µM lovastatin (LOV) (Sigma-Aldrich), based on previous protocols …” (line 916). Here, H₂Od is used as an abbreviation for distilled water, to maintain consistency across figures and text

Reviewer 2 Report
Comments and Suggestions for Authors
The paper "Mechanistic Insights into Cytokinin-Regulated Leaf Senescence in Barley: Genotype-Specific Responses in Physiology, Photosynthesis, and Protein Stability" aimed at identifying role of cytokinin hormones in induced leaf senescence in an important cereal crop, namely, barley. The study was carried out using a wide range of modern research methods, and the authors obtained a significant amount of experimental data. The manuscript is well structured, and the conclusions are well-founded and logical. However, I would like to make several recommendations to the authors for improving the presentation of the material and ask a few questions to clarify some details.
- I would recommend the authors slightly correct the title of the article. Actually, photosynthesis is one of the physiological processes occurring in plants, so the phrase "...in Physiology, Photosynthesis, and Protein Stability" is somewhat contradictory, as if it excludes photosynthesis from physiological processes. Most of the proteins studied by the authors are also directly related to photosynthesis. Perhaps the title should focus primarily on photosynthesis.
- Abstract. Sentences on lines 19-21 and 23-25 may be deleted, as these results are presented more generally in the sentence on lines 29-30. The sentence on lines 21-22 may also be deleted (at the authors' discretion), as the result presented there was not the key objective of the study. If the authors wish to retain the sentence on lines 19-21, then an explanation for the role of SAG12 protein in DIS is required. The abstract should include the results obtained for the Lomerit cultivar, because a brief description of the Bursztyn and Carina cultivars reactions are presented in the Abstract. In the abstract (line 15) should clarify that lovastatin was used to inhibit the mevalonate pathway of CKs biosynthesis; otherwise, the inclusion of lovastatin appears illogical. The full name of MEP should also be provided. The full name of the Methylerythritol Phosphate pathway is absent from the abstract, as well as throughout the manuscript.
- Line 61 in the Introduction should clarify that the ipt gene encodes isopentenyltransferase, which catalyzes one of the key reactions in cytokinin biosynthesis.
- Materials and Methods. The title of subsection 4.1 is "Plant Material, Growth Conditions, Leaf Senescence Induction, and Chemical Treatment." However, the description of chemical treatment is given in the following section, 4.2, on lines 842-844. The subsection titles should be changed.
- The description of chemical treatment includes references #67, 86-88. The accuracy of these references should be clarified, as references 86 and 87 do not contain BA and LOV treatment, and reference 88 is a review article.
- Section 4.7 should provide a brief description of the studied proteins: RCA, PsbO, PsbA, Lhcb1, Lhcb5, PsaB, Lhca4, SAG12, and eEF1a. The full names of many proteins and their functions are detailed in the discussion; however, it would be very useful to provide brief descriptions of them in the Materials and methods section. This will make it easier for interested readers to find information.
- Regarding the concentrations chosen for BA and LOV in the study, I ask the authors to clarify how exactly the concentrations of these compounds were selected and the effectiveness of BA and LOV treatment of barley leaves was determined. I would be particularly interested in how it was established that 40 μM lovastatin effectively inhibited cytokinin synthesis. Were endogenous cytokinin levels studied? If so, by what method? If not, are there plans to conduct further experiments to study endogenous cytokinin levels in barley leaves during DIS? This would be very useful in the context of this study. Moreover, on lines 772-775, the authors indicate that inhibition of the MVA pathway may enhance CK synthesis. It is possible that the effective concentration of LOV required to significantly reduce of MVA pathway activity varies among different cultivars.
- Lines 213-215. What do the authors mean by "cytokinin-mediated mitigation" – the possible involvement of endogenous cytokinins in DIS mitigation or BA-induced DIS mitigation?
- The abbreviation for lovastatin is given on line 88, but it is not always used in the text. The abbreviated version should be used throughout the text except for the first mention. A similar remark applies to the abbreviation CKs.
- Do the authors associate the intermediate position of Lomerit reactions with the fact that it refers to a winter barley? Has anyone compared the DIS process in different cultivars of winter and spring barley or other cereal crops?
- I would advise the authors to shorten the Conclusion slightly. It is currently very lengthy, taking up almost a page. I understand that the authors have obtained a great many results, but perhaps the authors will be able to summarize them a little more in the Conclusion, drawing the reader's attention to only the most important ones.
The comments made do not reduce the practical and fundamental value of the work and aimed only at improving the quality of presentation of the material.

Author Response
Manuscript ID: ijms-3872028
- Reviewer 2
All changes in the manuscript have been highlighted in yellow for clarity.
Comments 1: I would recommend the authors slightly correct the title of the article. Actually, photosynthesis is one of the physiological processes occurring in plants, so the phrase "...in Physiology, Photosynthesis, and Protein Stability" is somewhat contradictory, as if it excludes photosynthesis from physiological processes. Most of the proteins studied by the authors are also directly related to photosynthesis. Perhaps the title should focus primarily on photosynthesis;
Response 1: We thank the reviewer for this valuable suggestion. In response to the comment, we have revised the title to: “Mechanistic Insights into Cytokinin-Regulated Leaf Senescence in Barley: Genotype-Specific Responses in Physiology and Protein Stability”. This revised title better reflects the scope of the study without artificially separating photosynthesis from other physiological processes. At the same time, since our analyses also address ROS and cytokinin signalling, we believe that the use of the broader term physiology is more appropriate.
Comments 2: Abstract. Sentences on lines 19-21 and 23-25 may be deleted, as these results are presented more generally in the sentence on lines 29-30. The sentence on lines 21-22 may also be deleted (at the authors' discretion), as the result presented there was not the key objective of the study. If the authors wish to retain the sentence on lines 19-21, then an explanation for the role of SAG12 protein in DIS is required. The abstract should include the results obtained for the Lomerit cultivar, because a brief description of the Bursztyn and Carina cultivars reactions are presented in the Abstract. In the abstract (line 15) should clarify that lovastatin was used to inhibit the mevalonate pathway of CKs biosynthesis; otherwise, the inclusion of lovastatin appears illogical. The full name of MEP should also be provided. The full name of the Methylerythritol Phosphate pathway is absent from the abstract, as well as throughout the manuscript.;
Response 2: We thank the reviewer for these constructive comments, which helped us improve the clarity and completeness of the abstract. Following the suggestions:
- We have added an explicit explanation of the role of SAG12 protein, indicating its function as a cysteine protease associated with nitrogen remobilization and a marker of senescence onset.
- We have incorporated the results obtained for the Lomerit cultivar, describing its intermediate phenotype and responsiveness to cytokinin-related treatments.
- We have clarified that lovastatin was used to inhibit the mevalonate (MVA) pathway of cytokinin biosynthesis, thereby justifying its inclusion in the study.
- We have provided the full name of the MEP pathway (methylerythritol phosphate pathway) in the abstract and ensured consistency throughout the manuscript.
We believe that these revisions address the reviewer’s concerns and substantially improve the accuracy and readability of the abstract..
Comments 3: Line 61 in the Introduction should clarify that the ipt gene encodes isopentenyltransferase, which catalyzes one of the key reactions in cytokinin biosynthesis;
Response 3: We thank the reviewer for this valuable suggestion. Corrected sentence: “and by transgenic approaches, such as overexpression of the ipt gene, which encodes isopentenyltransferase, an enzyme catalyzing a key step in CK biosynthesis, under the control of the SAG12 promoter, activated at the onset of senescence, thereby prolonging leaf greenness through a precisely autoregulated loop” (line 63)
Comments 4: Materials and Methods. The title of subsection 4.1 is "Plant Material, Growth Conditions, Leaf Senescence Induction, and Chemical Treatment." However, the description of chemical treatment is given in the following section, 4.2, on lines 842-844. The subsection titles should be changed;
Response 4: We thank the Reviewer for this valuable suggestion. The corrected sentence is: the titles of the subsections are as follows: 4.1. Plant Material and Growth Conditions; 4.2. Leaf Senescence Induction and Chemical Treatment. (line 894)
Comments 5: The description of chemical treatment includes references #67, 86-88. The accuracy of these references should be clarified, as references 86 and 87 do not contain BA and LOV treatment, and reference 88 is a review article.;
Response 5: Thank you for this comment. Reference 86 (now renumbered as 88, due to the addition of new references) has been moved to earlier lines, as it relates to overall leaf sample preparation. References 87 and 88 (now renumbered as 89 and 90) were incorrectly cited — the correct citation should be (lines 917-919):
- Roman, H.; Girault, T.; Barbier, F.; Péron, T.; Brouard, N.; Pěnčík, A.; Novák, O.; Vian, A.; Sakr, S.; Lothier, J.; et al. Cy-tokinins are initial targets of light in the control of bud outgrowth. Plant Physiol. 2020, 172, 489–509. https://doi.org/10.1104/pp.16.00530
- Wang, W.; Hao, Q.; Tian, F.; Li, Q.; Wang, W. Cytokinin-Regulated Sucrose Metabolism in Stay-Green Wheat Phenotype. PLoS One 2016, 11(8), e0161351. https://doi.org/10.1371/journal.pone.0161351
Comments 6: Section 4.7 should provide a brief description of the studied proteins: RCA, PsbO, PsbA, Lhcb1, Lhcb5, PsaB, Lhca4, SAG12, and eEF1a. The full names of many proteins and their functions are detailed in the discussion; however, it would be very useful to provide brief descriptions of them in the Materials and methods section. This will make it easier for interested readers to find information;
Response 6: We thank the reviewer for this helpful comment. In the revised version of the manuscript, Section 4.7 (Materials and Methods) has been updated to include short descriptions of the studied proteins (RCA, PsbO, PsbA, Lhcb1, Lhcb5, PsaB, Lhca4, SAG12, and eEF1a). This section now provides their full names and concise information on their localization and function, which facilitates quick reference for readers without the need to search the Discussion (line 1027).
Comments 7: Regarding the concentrations chosen for BA and LOV in the study, I ask the authors to clarify how exactly the concentrations of these compounds were selected and the effectiveness of BA and LOV treatment of barley leaves was determined. I would be particularly interested in how it was established that 40 μM lovastatin effectively inhibited cytokinin synthesis. Were endogenous cytokinin levels studied? If so, by what method? If not, are there plans to conduct further experiments to study endogenous cytokinin levels in barley leaves during DIS? This would be very useful in the context of this study. Moreover, on lines 772-775, the authors indicate that inhibition of the MVA pathway may enhance CK synthesis. It is possible that the effective concentration of LOV required to significantly reduce of MVA pathway activity varies among different cultivars;
Response 7: We thank the reviewer for these valuable comments. The effective concentration of BA was determined in preliminary experiments in which different groups of exogenous cytokinins (kinetin, trans-zeatin, and benzyladenine) were tested in the range of 10⁻⁷ to 10⁻⁵ M. Based on these trials, BA was selected at a concentration that produced reproducible physiological effects without visible toxicity. However, these results are part of another manuscript.
For lovastatin, we based our selection on previously published data. In particular, the study by Wang et al. (2016, PLoS One, 11(8): e0161351, https://doi.org/10.1371/journal.pone.0161351) reported the use of lovastatin in cereal species as an inhibitor of cytokinin biosynthesis. In addition, another publication directly compared inhibitors of both the MVA and MEP pathways (lovastatin and clomazone, respectively) and applied higher concentrations of LOV (50 μM), where the physiological effects were comparable to those observed with 40 μM in our study (https://doi.org/10.18778/1730-2366.18.08).
Taken together, the selected concentrations were chosen to ensure effectiveness while maintaining consistency with existing literature. We agree that cultivar-specific differences in sensitivity may occur, and we have highlighted this point in the revised Discussion.
Comments 8: Lines 213-215. What do the authors mean by "cytokinin-mediated mitigation" – the possible involvement of endogenous cytokinins in DIS mitigation or BA-induced DIS mitigation?;
Response 8: We thank the reviewer for pointing out this ambiguity. In the revised version of the manuscript, the wording has been corrected. The phrase “cytokinin-mediated mitigation” has been replaced with “BA-induced DIS mitigation” to clearly indicate that the effect refers specifically to the treatment with exogenous benzyladenine, and not to the possible involvement of endogenous cytokinins. (line 218)
Comments 9: The abbreviation for lovastatin is given on line 88, but it is not always used in the text. The abbreviated version should be used throughout the text except for the first mention. A similar remark applies to the abbreviation CKs.;
Response 9: We thank the reviewer for this remark. In the revised manuscript, the abbreviations for lovastatin (LOV) and cytokinins (CKs) are now used consistently throughout the text, with the full names provided only at their first mention.
Comments 10: Do the authors associate the intermediate position of Lomerit reactions with the fact that it refers to a winter barley? Has anyone compared the DIS process in different cultivars of winter and spring barley or other cereal crops?;
Response 10: The barley cultivars were selected based on their distinct patterns of senescence onset. The spring cultivar Carina and the winter cultivar Lomerit have been employed in previous studies (https://doi.org/10.1111/j.1399-3054.2011.01545.x; https://doi.org/10.1093/pcp/pcaa114), where they were shown to exhibit different senescence dynamics, despite both following the onset of natural senescence. At the same time, other authors have successfully applied the DIS (or DILS) method to precisely control the initiation of senescence (https://doi.org/10.1007/s00726-014-1912-y), which is particularly useful when comparing cultivars of different growth habits, such as winter and spring barley. In our study, special emphasis was placed on the newly analyzed cultivar Bursztyn, which unexpectedly exhibited a delayed senescence phenotype that, to our knowledge, has not been previously described. The use of Carina and Lomerit as well-established reference cultivars also allowed us to better understand the senescence mechanism in Bursztyn.
Comments 11: I would advise the authors to shorten the Conclusion slightly. It is currently very lengthy, taking up almost a page. I understand that the authors have obtained a great many results, but perhaps the authors will be able to summarize them a little more in the Conclusion, drawing the reader's attention to only the most important ones.;
Response 11: We thank the Reviewer for this valuable comment. In line with the recommendation, the Conclusion section has been revised to eliminate redundancies and its length has been reduced by approximately half. The revised version now provides a concise synthesis of the main findings, emphasizing the key cultivar-specific senescence patterns and cytokinin responses, while avoiding unnecessary repetition of results.

Round 2
Reviewer 1 Report
Comments and Suggestions for Authors
Thanks for authors’ works, most comments were well addressed. The mRNA data and other data could be cited in manuscript as personal communication, which would fill gaps between biochemical, and protein data. Good luck.
Author Response
Comment: Thanks for authors’ works, most comments were well addressed. The mRNA data and other data could be cited in manuscript as personal communication, which would fill gaps between biochemical, and protein data. Good luck.
Response: We appreciate the Reviewer’s valuable suggestion. In accordance with this comment, we have included in the revised version of the manuscript an additional note stating that “future analyses of the barley cultivar Bursztyn should also address changes in the expression of chlorophyll metabolism–related genes as well as the dynamics of HmChl a accumulation, especially since total Chl a+b content was maintained at comparable levels in BA-treated and untreated samples under DIS.”
However, since the manuscript, including the mRNA expression analyses, is currently under review, we are, at this stage, limited to referring to these data in a general form.
Reviewer 2 Report
Comments and Suggestions for Authors
I am grateful to the authors for their prompt and high-quality revision of the manuscript. All my suggestions and recommendations were taken into account, the necessary corrections were made, and I received comprehensive answers to all my questions. I have no further comments.
Author Response
Comment: I am grateful to the authors for their prompt and high-quality revision of the manuscript. All my suggestions and recommendations were taken into account, the necessary corrections were made, and I received comprehensive answers to all my questions. I have no further comments.
Response: We sincerely thank the Reviewer for the positive evaluation of our work and the appreciation of our revisions. We are grateful for the valuable comments and suggestions provided during the review process, which have helped us to improve the quality and clarity of the manuscript.